# Meat Analogues: Relating Structure to Texture and Sensory Perception

**DOI:** 10.3390/foods11152227

**Published:** 2022-07-26

**Authors:** Layla Godschalk-Broers, Guido Sala, Elke Scholten

**Affiliations:** Physics and Physical Chemistry of Foods, Wageningen University & Research, P.O. Box 17, 6700 AA Wageningen, The Netherlands; layla.broers@gmail.com (L.G.-B.); guido.sala@wur.nl (G.S.)

**Keywords:** meat analogues, sensory test, TPA, microstructure

## Abstract

The transition from animal to plant proteins is booming, and the development of meat analogues or alternatives quickly progressing. However, the acceptance of meat analogues by consumers is still limited, mainly due to disappointing organoleptic properties of these foods. The objective of this study was to investigate possible relationships among structure, textural characteristics, consumer acceptance, and sensory evaluation of commercially available meat analogues. The microstructure and texture of 13 chicken analogue pieces and 14 analogue burgers were evaluated with confocal laser scanning microscopy (CLSM) and texture profile analysis (TPA). The moisture of the samples was related to cooking losses and release of liquid upon compression after cooking. Meat products were included as references. A sensory panel (*n* = 71) evaluated both flavour and texture characteristics. For the chicken analogue pieces, samples with more added fibres had a harder and chewier texture but were less cohesive. No other relations between composition and structure/texture could be found. In the sensory evaluation, lower hardness and chewiness were only seen in products with more fat. A lower sensory hardness was found to be related to the presence of small air pockets. For analogue burgers, there was no clear relation between composition and structure/texture. However, instrumentally measured hardness, chewiness, and cohesiveness correlated well with the corresponding sensory attributes, even though they could not be clearly linked to a structural feature. Next to this, fat content showed a clear correlation to perceived fattiness. CLSM images of burgers with high perceived fattiness showed large areas of fat. Therefore, the release of large fat pools from the meat was most likely responsible for the perception of this attribute. However, perceived fattiness was not related to liking, which was the case also for chicken analogue pieces. For both pieces and burgers, even if some of the measured textural attributes could be linked to the sensory profile, the textural attributes in question could not explain the liking scores. Liking was related to other aspects, such as meaty flavour and juiciness, which were not directly linked to compositional or textural features. Juiciness was not directly related to the moisture loss of the products, indicating that this attribute is rather complex and probably involves a combination of characteristics. These results show that to increase the appreciation of meat analogues by consumers, improving simple texture attributes is not sufficient. Controlling sensory attributes with complex cross-modal perception is probably more important.

## 1. Introduction

The number of meat analogues available for consumers is growing worldwide, facilitated by innovation, a greater number of product launches, and an increased willingness of consumers to buy these products [1,2]. With an increasing world population and a growing concern for animal welfare and environmental sustainability, the need for the transition from an animal-based to a more plant-based diet is increasing [3,4,5,6]. Currently, the main protein sources in our diets are still animal based [7]. Despite the growing availability of meat analogues, the transition towards plant-based diets is still hampered, as the sensory properties of the available products do not sufficiently resemble those of meat. To replace meat with meat analogues, similar sensory properties are a prerequisite for acceptance [8,9,10].

In recent years, considerable research has been performed to develop meat analogues with improved quality, aiming for similar texture, nutritional values, and taste. Different technologies are used to structure plant proteins, such as extrusion, shear cell technology, wet or electrospinning, and culturing [11,12,13,14]. These technologies provide solutions to obtain meatlike structure and texture in meat analogues [15,16,17,18,19]. However, studies on the relationship between structural characteristics of meat analogues and sensory perception are scarce. More research is required to identify the aspects of meat analogues related to sensory perception and consumer appreciation. One of the few studies that included structure, physical properties, and sensory characteristics of meat analogues was performed by Chiang and coworkers (2019), who observed that textured soy protein isolate formulations with either 20% or 30% wheat gluten could resemble chicken meat texture relatively well in terms of measured hardness and chewiness. However, the sensory hardness and chewiness of these products were significantly higher than those of cooked chicken breast. The sensory profile of meat analogues may not be easily explained by specific textural attributes, and to develop higher-quality meat analogues, more knowledge is required to identify possible relationships between structural characteristics and specific sensory attributes.

The aims of this study were to (a) unveil possible relations among microstructure, texture characteristics, and sensory perception of commercially available meat analogues and (b) identify drivers for consumer acceptance. From the vast number of available meat replacers, two representative categories were selected, chicken pieces and burger patties. For each category, products with different types of protein were selected to have samples with different textures and flavours. Meat reference products (i.e., chicken fillet and beef burger) were included in the tests. Microstructure analysis was performed with confocal microscopy, and moisture loss was determined after cooking. Texture profile analysis (TPA) was used to measure different texture attributes relevant for meat analogues.

## 2. Materials and Methods

### 2.1. Sample Selection and Preparation

The Innova Insights database was used to identify meat analogues marketed in the period 2014–2019, yielding over 8000 results. Products mimicking chicken pieces and beef burgers were chosen as representative product types. Only products with a minimum energy percentage from protein of 12% were chosen, which is set as a requirement for suitable meat replacers by the Netherlands Nutrition Centre (Voedingscentrum), and products in which protein was the main ingredient. Selection was based on availability and variety in composition with respect to protein type, total fat content, and fibre. Detailed information on the composition and the nutritional values can be found in Appendix A.

Samples were stored frozen (−20 °C) until use. They were thawed at 4 °C 24 h prior to preparation and accompanying measurements. Cooking of samples was performed using a regular frying pan on a gas stove with a pan temperature at 200 °C. Sunflower oil (Jumbo) was used as cooking fat (15 g oil/100 g product). All samples were cooked until a core temperature of 74 °C was reached. The prepared products were left to cool down before performing additional measurements. All measurements for physical characterization and sensory evaluation were performed using the cooked products.

### 2.2. Physical Characterization

#### 2.2.1. Moisture Content and Expressible Moisture

The moisture content (MC) of the cooked samples was determined according to the AOAC method 950.46 [20]. Samples were dried at 105 °C in an oven (Memmert, Büchenbach, Germany) for 16–18 h, until constant weight was obtained, and the results were expressed as a percentage of wet weight.

The expressible moisture (EM) of the cooked samples was measured by compressing the cooked samples between 2 filter papers (Whatman no. 4) and determining the moisture absorbed by the filter papers [21,22]. A TA.XT Plus Texture Analyser (Stable Micro Systems, Surrey, UK) equipped with a 75 mm flat-end probe and a 50 kg load cell was used for the compression. Test conditions included 60% deformation of the sample and a compression speed of 0.8 mm/s. All measurements were performed in triplicate. The expressible moisture (EM) was expressed as a percentage of the initial sample weight. The specific contributions of water and fat to the expressible moisture were determined by drying the filter papers in an oven (Memmert, Büchenbach, Germany) at 105 °C overnight until constant weight, assuming complete evaporation of water.

#### 2.2.2. Cooking Loss and Fat Absorption

Cooking loss (CL) was determined by preparing the samples in a dry pan, as described previously, and calculating the weight loss in percentage (%). All measurements were performed in triplicate.

Fat absorption (in g oil/100 g sample) was determined by cooking the samples in oil (15 g/100 g sample) as described previously. By determining the difference in weight before and after cooking and taking into account the water loss during cooking, fat absorption (*FA*) was calculated as
(1)$$FA\;\left(\frac{g}{100g}\right)=weight\;after\;cooking-initial\;weight-cooking\;loss$$

#### 2.2.3. pH

The pH was determined in triplicate on a blended homogenate of the samples using a Metrohm 836 pH Mobile (Barendrecht, The Netherlands) with a glass electrode.

### 2.3. Texture Profile Analysis (TPA)

Texture profile analysis (TPA) was performed using a TA.XT Plus Texture Analyser (Stable Micro Systems, Surrey, UK) to determine hardness, springiness, cohesiveness, and chewiness. A 75 mm flat-end probe with a 50 kg load cell was used for the compression. Samples were cooked prior to the analysis (see Section 2.1) and cooled to room temperature. Although the size of the samples was kept as much as possible the same, slight variations in thickness and size were unavoidable. The specific thickness and surface area were measured for each sample and were used for further calculations. For each sample, measurements were performed in triplicate as a double compression test with 50% deformation, with 5 s between the two compression steps and a compression speed of 2 mm/s. Textural attributes were extracted from the stress–strain curve [23,24,25]. Hardness was defined as the peak tensile stress (kPa) (P1) in the first compression cycle (50% compression), springiness was defined as the rate at which the deformed material returned to its original state (D2/D1), cohesiveness was determined from the area of the first compression and the second compression peak ((A1–A2)/(B1–B2)), and chewiness was taken from the peak stress and the areas and time of the first and second compression (P1×(B1/A1)×(D2/D1)). An overview of the test and the corresponding indications of the letters is given as Appendix A.

### 2.4. Confocal Laser Scanning Microscopy

Microstructure images were obtained using a confocal laser scanning microscope (CLSM) (Zeiss LSM 510 META, Breda, The Netherlands). A representative selection of samples (the best and lowest scoring samples from the sensory test, and some in between) was prepared by pan frying (3–4 min), as described in Section 2.1. Thin slices (~300 µm) were taken from the inner part of the samples using a scalpel. For samples with a strong fibrous character, slices were taken both along and against the grain. Slices were put on microscopic glass slides and topped with a 250 µm thick gene frame spacer (15 × 16 mm, Thermo Fisher Scientific SAS, France). Three microlitres of a fluorescent dye solution containing Alexa Fluor^®^ 488 NHS (0.05% (*w*/*v*)) (for protein staining, green) and Bodipy^®^ 665/676 (0.05% (*w*/*v*)) (for fat staining, red) in dimethyl sulfoxide (DMSO) were poured on each sample slice and left for 10 min to allow redistribution within the sample. Excess dye was removed using filter paper, and samples were covered with a coverslip. Dyes were excited at 488 nm (Ar) and 633 nm (HeNe), and the fluorescence emission recorded in the wavelength ranges of 400–565 nm and 600–700 nm, respectively. Images of representative areas were taken using 63× Plan-Apochromat (Zeiss) and 10× Plan-Neofluar (Zeiss) magnification objectives.

### 2.5. Sensory Evaluation

Quantitative sensory analysis was performed using a nontrained consumer panel (*n* = 71, 63% female, 37% male, aged 19–41 years, and nonvegetarians). The analysis of the samples was performed at the participants’ homes. The participants were instructed to perform the tasting in a neutral environment (quiet, no other smells) and to use separate trays and forks for each sample. All samples were delivered frozen to the participants in a bag containing all necessary products and information. The bag contained chicken and chicken analogue pieces and (smaller portions of) beef burger and analogue burger samples in anonymous vacuum-sealed plastic bags. Sample bags were labelled with random 3-digit codes. Paper booklets with the questionnaires, instructions, napkins, crackers for taste neutralization, numbered plastic trays, plastic drinking cups, plastic forks, and a bottle of cooking oil (sunflower oil) were also provided. Samples were stored and prepared by the participants according to the provided guidelines. Despite providing cooking instructions, some differences in cooking time and temperature may have caused differences in hardness, dryness, or colour. Even so, this method provides a more accurate view on opinions on the preparation of products by consumers than tests in a tasting lab [26]. Then, the environment of the participants likely resulted in a sense of familiarity and calmness, supporting scores more representative of real-life consumption [27]. Prior to the tastings, the participants were informed about the type of products (meat replacers) used in the test, and asked questions about their eating habits and attitudes regarding meat replacers. They were then asked to taste the different samples and fill in the questionnaire about the samples in a random order. The participants completed the evaluation of all the samples within 2 weeks.

The questionnaire contained questions about the raw product (e.g., look and smell) and the preparation of the product (e.g., duration and ease) and hedonic and descriptive questions about the cooked product (e.g., hardness and chewiness). For the hedonic questions, a 9-point hedonic scale was used; this scale has been reported to be reliable and have a high stability of response [28]. For the descriptive questions, a 0–100 VAS scale was used, as this scale allows for a high sensitivity of differences as panellists are not restricted by limited options [29]. A description of the attributes was provided (see Table 1).

### 2.6. Statistical Analysis

Statistical analysis was carried out using IBM SPSS software version 25 (IBM Corporation, New York, NY, USA, 2017). Comparisons were made using one-way analysis of variance (ANOVA), followed by Tukey’s HSD pairwise comparisons of means. A post hoc comparison of meat vs. meat replacers was performed with Dunnett’s test. Multiple stepwise linear regression was used to identify relationships between the dependent and independent variables for liking. The degree of association between variables was determined using partial correlation. Observed differences were considered significant at *p* < 0.05. To determine Pearson correlation coefficients between measured textural attributes and sensory attributes, both the sensory and instrumental data were averaged for each of the investigated attributes. The sensory data were further analysed using principal component analysis (PCA) to identify factors that can explain most of the variance between the meat analogues. PCA was carried out using XLSTAT software (XLSTAT, 2022, Addinsoft, New York, NY, USA).

## 3. Results and Discussion

### 3.1. Composition

To be able to understand the sensory perception of meat analogues in terms of their composition and structure, we selected a variety of chicken analogue pieces and beef burger analogues (Table 2). Most samples in this study were based on protein from soy, wheat, pea, mycoprotein, or combination thereof (Table 2). The amount of protein was on average lower in both chicken pieces and burger analogues (20% and 17%, respectively) than in real chicken meat (24%) and beef burgers (20%), with some exceptions. In chicken analogue pieces, the highest protein content was found in products containing pea protein (*r* = 0.704, *p* = 0.016), such as Gold & Green (30%), AH (23%), and Naturli (21%), but also in products based on soy protein and wheat (VegiDeli Chicken Style Pieces) (26.6%). However, another product with a combination of soy and wheat (Vantastic Foods) had the lowest protein content (15%). A similar low-protein content of 15.3% was also seen for the product made of mycoprotein (Quorn). In the case of burger analogues, the products with the highest protein content were those produced with a combination of wheat, pea, and mycoprotein (21%, Quorn) and one consisting of soy and wheat (20%, AH). A product based on a combination of soy and wheat had the lowest protein content (14%, Fry’s). Therefore, for both the chicken pieces and burger analogues, the total protein content is more dependent on total composition than on the protein source.

The protein source seemed to have an influence on the appearance of the burgers. As a matter of fact, for the uncooked samples (Appendix A), it was possible to discriminate between a ‘precooked’ and a ‘raw’ appearance. All burgers with a ‘precooked’ appearance were based on soy and/or wheat. The burgers with a more ‘raw’ appearance were mostly based on pea only or a combination of other components with pea. Only two of these six samples did not contain any pea and contained mostly soy. This suggests that a more wet, ‘raw’ look can be more easily obtained using pea protein. Meat replacers that look like raw meat before cooking are supposedly more readily accepted due to their likeness to real meat [9,30]. In our study, however, the raw appearance alone did not affect liking.

For both chicken analogue pieces and analogue burgers, a large variation in fat content was observed. For chicken analogue pieces, fat ranged from 0.4% to 11.8%, whereas real chicken contained about 2% fat. For analogue burgers, fat ranged from 5.2 to 19%, whereas real beef burgers contained 17% fat. From the ingredients list, it could be seen that different types of oil and fat were present, such as sunflower, rapeseed, coconut, and palm oil.

Some correlations were observed concerning the main protein type of the meat analogues and the fat content. A higher amount of fat was present in chicken analogue pieces with wheat as main protein source (*r* = 0.847, *p* = 0.001) and in analogue burgers based on pea (*r* = 0.661, *p* = 0.027). Burgers based on soy had a lower fat percentage (*r* = −0.730, *p* = 0.011). Although no direct link between protein type and fat content is expected, this may be related to the water-binding capacity of soy proteins [31,32], which is relatively high compared with other plant-based proteins. Higher water binding may allow for less addition of fat to achieve a higher sensory juiciness. The amount of added fibres, such as methylcellulose, wheat, or pea fibre, varied in the different meat analogues. These fibres are often added for their water-binding properties [3]. In chicken analogue pieces, fibre contents ranged from 0.2 to 7.6 g/100 g. In analogue burgers, it ranged from 0 to 7.5 g/100 g (Table 2).

The average moisture content (MC) of cooked samples ranged between 33% and 66% and was not significantly different between chicken pieces and burgers, or between meat and meat analogues (Table 2). In general, soy-based products showed a higher MC than other protein sources. The higher water content is most likely related to the higher water-holding ability of soy in comparison with that of wheat and other proteins [33,34]. For example, Oumph the Chunk (Food for Progress) showed an MC of 64.94% ± 2.72, which matched the MC of chicken (65.7% ± 1.80). Similar findings were seen in the case of analogue burgers, where soy-based samples had a higher MC (56% on average, *r* = 0.538, *p* = 0.001) than those with other proteins. Products based solely on wheat or wheat combined with pea protein had the lowest MC. Additionally, in this case, the products containing soy matched the MC of real beef (59.29% ± 0.62) the most, but also products containing a combination of soy and wheat or pea protein resembled the MC of real beef burgers. Next to the water-holding capacity of the protein, differences in MC can also be related to processing conditions. Production parameters, such as the screw speed and temperature during extrusion, have been shown to affect MC [35]. Furthermore, variations in initial moisture content before extrusion will affect the final MC. As the exact processing conditions of these commercial samples are not known, it is not clear whether the differences in MC are a result of the different functionality of the protein source or an effect of different processing conditions.

### 3.2. Cooking Loss and Expressible Moisture

Cooking loss (CL) is the moisture released during cooking of the product up to a standardized core temperature. Both real chicken pieces (23% ± 1.6) and burgers (25% ± 2.5) had the highest CL, which was on average 10% higher than that of the meat analogues (Table 3), also according to the literature (18.8%–35.8%) [36]. A higher MC in the uncooked samples was related to greater CL in both chicken pieces (*r* = 0.53, *p* = 0.08) and burger samples (*r* = 0.60, *p* = 0.03). Cooking loss ranged from 7% ± 0.3 to 19% ± 1.4 in chicken analogue pieces, and from 4% ± 0.48 to 23% ± 1.2 in analogue burgers, both giving an average of 13%. Differences in CL can be expected to be related to a variety of different parameters, such as the structure of the protein network, the type of the protein, the water-holding ability of the proteins, and the fibres [37]. In previous studies, confined compression tests showed a relationship between the pore structure of meat analogues and CL [38]. These cavities may provide easy release of water from the products [39,40]. In our study, products with large and/or many air cavities observed using CLSM (Figure 1) did not have a greater CL than others. For example, Vivera—Chicken Pieces (Figure 1G2) and Linda McCartney—Quarter Pounder (Figure 1J2) showed large pore sizes but did not give higher cooking losses. Instead, Valess—Fillet Pieces and Quorn—Chicken Pieces showed the highest values of CL of 18% and 16%, respectively. Although Valess also showed larger pore sizes, Quorn showed a dense structure without a lot of pores. In the case of analogue burgers, Beyond Meat showed the highest CL, but also in this case, no evidence of large pores was observed (Figure 1H2). In addition, no relation could be found between CL and other compositional factors. CL may thus be more related to specific features due to other compositional factors (i.e., emulsifiers, stabilizers, processing conditions, ingredient interactions, and the properties of the formed protein network) [41,42,43]. It may be expected that with a higher release of juices during cooking, sensory juiciness will be reduced, as limited moisture will remain in the cooked products.

Moisture or fluid loss during mastication is expected to play a large role in the sensory evaluation of meat analogues. As a matter of fact, this fluid stimulates the receptors in the mouth and is presumably related to sensory juiciness [22,44,45,46,47]. To gain insight into the moisture loss of the different samples, we measured the expressible moisture (EM) during compression of the cooked samples. EM (%) was highest in meat samples (chicken, 3.7% ± 0.95; beef, 1.7% ± 0.21) and significantly lower in all meat analogues (Table 3). Meat samples also had the highest cooking loss during preparation. Therefore, even though more fluid was already released during cooking, they were still able to release more fluid during compression. This could be related to specific structural and biochemical changes occurring in the muscle tissue in real meat. EM values in chicken analogue pieces ranged from 0.37% to 1.4% (average: 0.84%), and in burger analogue samples from 0.35% to 0.99% (average: 0.58%). Therefore, even though large differences were observed for CL, there were limited variations in EM. This could indicate that EM is just determined by protein type and structure but is not related to the initial moisture content. The MC of the real meat samples was not significantly higher than that of the meat analogue samples (Table 2). This indicates that upon compression, moisture is more readily released from real meat. Several studies have established that heat exposure during processing of meat analogues induces structural changes that favour higher water-holding ability [33,40,48]. This may explain why EM in meat analogues was lower. In addition, fibres are often added to increase the water-holding capacity of the products. More fibres resulted in a higher MC (*r* = 0.405, *p* = 0.013) for the chicken analogue pieces and a higher cooking loss (*r* = 0.361, *p* = 0.028). Therefore, initially, these products were able to hold more water, but this was released again during preparation. As most water was already released, these samples did not exhibit a higher EM (*p* = 0.416).

Despite standardization of the cooking method, fat absorption (Table 3) varied largely among samples. Fat absorption during cooking varied between −18% and 24% for the chicken analogue pieces (average: −0.7%) and between −9% and 5% for the burger analogues (average: −0.6%). Negative values indicate a greater fat loss rather than oil uptake. The average values of the two types of samples were comparable to the fat absorption of chicken and beef during cooking (0.04 and −1.0%, respectively). Although the effect of CL was taken into account in this calculation, we cannot be sure that this was only oil loss. Additionally, water may be released. As CL was determined in a dry pan, this may not be completely representative of the CL in the presence of oil.

The fat released in the EM did not show a clear relation with the initial fat content of the samples, as the fat lost upon compression included both the fat initially present in the samples and the fat adsorbed during cooking. Therefore, differences among samples were also related to the ability of the samples to absorb oil during preparation. EM in real chicken meat appeared to be mainly water, while fat remained present in meat. This was likely due to the low fat content of chicken breast and the limited oil absorption during frying. On the other hand, the chicken analogue pieces, having a higher fat content and having in some cases absorbed a significant amount of oil during preparation, expressed more fat together with water. In chicken analogue pieces, around 40%–80% (although not all significant) of the released moisture (EM) consisted of fat, whereas for real chicken, this was only 18% (Table 3). For burgers, the fat content in EM was between 69% and 82% for the analogues, and only 42% in the real beef burger (Table 3). The higher values for the burgers could be explained by the higher initial fat content of the samples (*r* = 0.436, *p* = 0.007). The initially present fat may be more enclosed within the structure, and may not be released easily, whereas the adsorbed fat may be present closer to the surface of the product, and can therefore be released more readily. With respect to the absorbed fat, samples with the highest oil absorption during cooking (Naturli Chick Free, 24%) did not express a higher amount of oil in the released moisture. As the amount and composition of the released moisture did not show a clear relation with either the original fat content (Table 2) or the fat absorption during cooking (Table 3), the released moisture may be more related to other characteristics, such as protein type, protein network, and water and oil-binding capacity of the systems.

### 3.3. Structure of the Meat Analogues

To gain insights into differences in structure among the studied meat analogues, we used microscopy to visualize the fat phase (in red) and the morphology of the protein network (in green). By taking samples from the middle of the products (as opposed to sampling from the edge), we assume that the images reliably reflect the microstructure. In Figure 1, pictures of the general appearance of a selection of meat analogues are provided, together with CLSM micrographs. The latter clearly show for all samples fibrouslike or layered structures, as well as randomly oriented coarsely connected protein networks. For example, a fibrouslike structure is clearly visible in AH Pieces Like Chicken, made of wheat protein (Figure 1A). This type of structure is most likely obtained by extrusion [39]. Wheat gluten has been reported to enhance fibrous structural arrangements in meat analogues [19,49]. Similar structured domains were also observed for Vantastic Foods pieces, which also contain wheat protein (Figure 1E). However, also pieces without wheat showed fibrouslike structures, such as the Greenway Chick Pieces (Figure 1B), which consisted of soy only. These fibrous structures can easily be seen in the images in Figure 1B1. Veggie Chef (Figure 1F) and Vivera Pieces (Figure 1G) showed limited fibrelike structures and a more course protein network, even though they also consisted mainly of soy. A very interesting image was obtained for Quorn Pieces (Figure 1C), revealing a homogeneous distribution of very small mycoprotein fibres. The protein structure of meat analogues is thus determined not only by the composition, but also by the specific processing conditions [14,17,39]. The fat phase of the analogue pieces was distributed as small droplets throughout the samples, except for AH (A) and Veggie Chef (F).

In the burgers, we could also distinguish some differences in structure with respect to the protein phase and the fat phase. Two of the burger analogue samples showed clearly aligned fibrouslike protein domains: the Vegetarische Slager—Mc2 Burger and the VegiDeli Burger (Figure 1K,L). Just like with the pieces, the structure was not directly related to the protein source: both burgers were soy based, but only the Vegetarische Slager sample contained gluten. Most other samples, which were based on pea or soy protein with or without gluten, did not show clear fibrouslike structures, but a more homogeneous protein network. The structure of these analogues was much less fibrous than of real beef burgers, which commonly contain fibre bundles up to 50%–70% in an isotropic protein network [50]. As analogue burgers contained more fat than the chicken analogue pieces, more differences in the fat distribution could be observed. Both small fat droplets and large fat pools were present in analogue burgers. However, as the structure of the analogue burgers was very inhomogeneous, both small and large fat areas could be seen in the same product. For example, in the Vegetarische Slager—Mc2 Burger (Figure 1K) and Beyond Meat—Beyond Burger (Figure 1H) samples, images revealed both large pools of oil and smaller randomly distributed fat globules. The other samples showed a more even distribution of fat droplets, although some fat droplets appeared as single droplets and some more in an aggregated form. The oil distribution did not show much correlation with the amount of fat absorbed or lost during cooking. For example, the samples with the largest pools (i.e., Vegetarische Slager—Mc2 Burger (Figure 1K) and Beyond Meat—Beyond Burger (Figure 1H)) showed an oil absorption of +1.88% and −5.31%, respectively.

### 3.4. Texture Analysis

The results of the measured textural attributes can be found in Table 3. For both chicken analogue pieces and analogue burgers, hardness values varied substantially. That of chicken analogue pieces ranged between 24 and 165 kPa, and most samples showed a much lower value than the real chicken (142 kPa ± 118). Higher values for hardness indicate a texture that is more ‘tough’ [51]. No clear relation could be seen between hardness and the general structural organisation, as seen in CLSM pictures. However, samples with high hardness, such as Veggie Chef and Quorn, had a compact dense structure. On the other hand, also Vivera and Greenway showed high hardness, but these samples had many air pockets present or a clear fibrous structure, respectively. For Vantastic Foods, we also saw a dense structure, but this sample showed very low hardness. This is likely related to the high number of well-defined small air bubbles that were enclosed in the dense structure and a homogeneous distribution of the fat globules. Both air bubbles and fat globules may act as structure breakers, leading to lower hardness. For springiness, the values of the chicken analogue pieces (ranging between 0.65 and 0.91) were similar to those of real chicken (0.69 ± 0.03). No clear relation between springiness and structure could be seen. For example, slightly higher values of springiness were found for samples with a fibrous structure (as AH) and for samples without this property (Veggie Chef and Vivera). Additionally, for cohesiveness, no clear relation with the structure could be observed, and in general, the chicken analogue pieces showed greater cohesiveness (range: 0.36–0.66; average: 0.51 ± 0.11) than real chicken (0.40 ± 0.04). As for chewiness, similar results were observed in relation to the structure: high values of chewiness (87–142) were independent of fibrosity. However, dense structures present in Vantastic Foods gave very low values of chewiness (16 kPa) in comparison with real chicken (51 kPa). Textural attributes, therefore, did not seem to be strongly related to the fibrous nature of the samples.

For chicken analogue pieces, the presence of added fibres correlated positively with TPA values for hardness (*r* = 0.523, *p* = 0.001) and chewiness (*r* = 0.357, *p* = 0.030), while it was negatively correlated with cohesiveness (*r* = −0.439, *p* = 0.007). No clear relation was seen between textural attributes and other parameters, such as moisture content. The hardness, springiness, cohesiveness, and chewiness of chicken analogue pieces of the brands Valess and de Vegetarische Slager were most similar to those of real chicken.

Analogue burgers exhibited hardness values ranging from 70 kPa (Vegafit) and 395 kPa (SoPeace), whereas real beef burgers showed a hardness of 210 kPa. Neither the fibrous structure nor the distribution of fat could be related to hardness. In burgers, a higher amount of fibres was associated with lower EM (*r* = −0.411, *p* = 0.012). This may be related to different water-holding properties of the fibres in burgers. Burgers were all comparable in springiness (range: 0.40–0.89; average: 0.68 ± 0.14). Cohesiveness ranged between 0.16 and 0.51 (average: 0.36 ± 0.10), and chewiness between 9 and 194 kPa (average: 67 kPa ± 54). Quorn Supreme Burgers were most similar to beef burgers for all attributes, as well as Garden Gourmet and AH Burger Deluxe.

These results show that it is difficult to relate textural attributes to specific features of the meat analogues. However, these results point out that the denseness of the structure, and the inclusion of fat, air, and fibrous structures, may be important characteristics.

### 3.5. Sensory Profile

To characterise the sensory profile of the studied samples, we investigated different aspects: (1) appearance and ease of preparing, (2) relevant sensory attributes, and (3) liking.

#### 3.5.1. Evaluation of Raw Meat Analogues

Table 4 shows the scores of the panel members for the attractiveness of the raw, uncooked samples, which were based on the evaluation of appearance, smell, and colour. Appearance and especially colour give a first impression of the product and may already determine whether consumers will buy a certain meat analogue [52,53,54,55]. The burger samples could be clearly separated into a raw meat-look and a ‘precooked’ appearance. According to the scores of the panel members, the raw look was more appealing and more meatlike. The only exception was the AH Burger Deluxe, which also received a high score even though it had a more ‘precooked’ appearance (Appendix A).

The doneness of meat during cooking is often assessed by the change in colour [56]. Some of the analogue burger samples in this study retained their red colour after cooking, which resulted in negative responses, such as ‘Looks too red/raw after cooking’, ‘Artificial/fake colour’, ‘Takes too long to cook’, and ‘I don’t know when it’s cooked’ (Beyond Burger, Garden Gourmet Incredible Burger, Quorn Supreme Burger; data not shown). This effect is explained by the presence of colourants. Heat-labile colourants (e.g., beetroot) are often added to meat analogues to resemble the appearance of raw meat [2,57]. Depending on the amount of colouring and colour degradation time, a pink look, associated with rawness, can still be visible during cooking. A too long persistence of a pink look may lead to prolonged cooking time and loss of food quality, and may result in lower consumer appreciation [56].

The ratings for ‘preparation is easy’ and ‘preparation takes long’ for the meat analogues were similar to those of the real meat, and for analogues, even lower values for ‘preparation takes long’ could be seen. The average scores (1–5) for ease of preparation in the ranges of 3.43–4.16 (chicken analogue pieces) and 3.23–4.07 (analogue burgers) were given, compared with the scores of 3.75 and 3.56 for the real meat products, respectively, indicating that all samples were moderately to very easy to cook (Table 4), even though some participants mentioned problems such as meat analogue samples falling apart or sticking to the pan. More differences were visible for ‘preparation like meat’, where meat analogues scored significantly lower than the real meat samples.

#### 3.5.2. Attributes and Liking

The results of the sensory tests with respect to the sensory attributes and liking can be found in Table 5 and Table 6. Compared with real chicken, chicken analogue pieces generally scored higher on colour (darker) and fattiness, and lower on hardness, cohesiveness, fibrousness, and meaty flavour (differences were not significant for all samples). In the case of chewiness, juiciness, and flavour intensity, both lower and higher scores for the chicken analogue pieces were observed in comparison with real chicken. According to the sensory panel, several products were not significantly different in the textural attributes (hardness, chewiness, cohesiveness, fibrousness, juiciness, and fattiness) compared with chicken (Vivera, Veggie Chef, Greenway, AH—Like Chicken, and Food for Progress), whereas others differed in all or almost all those attributes (Quorn and Gold & Green).

Compared with real beef, analogue burgers were generally perceived as less chewy, cohesive, fibrous, juicy, fatty, and with a lower flavour intensity and meaty flavour. From a statistical point of view, the scores of most attributes were not significantly different from those of the reference, except juiciness and fattiness. Only for the Moving Mountains Burger, juiciness and fattiness were similar to those of the real beef burger. The Quorn Burger matched the real beef burger the most on hardness, chewiness, cohesiveness, and fibrousness, followed by the burgers from Vivera, Gardein, and Linda McCartney.

For both chicken analogue pieces and analogue burgers, a close resemblance of all textural attributes did not yield a liking score equal to that of real meat (Table 6). Stepwise linear regression analysis including all attributes showed that meaty flavour and juiciness could explain liking for 48% in chicken analogue pieces (B = 0.589 and 0.240, respectively) and 47% in analogue burgers (B = 0.572 and 0.236, respectively). Other mouthfeel attributes (flavour intensity and fattiness) and textural attributes (hardness, chewiness, etc.) were not predictive for liking (either overall liking or liking of texture, flavour, or appearance). These findings were visualized using PCA plots (Figure 2). The position of the tested products in the graphs (in blue) relates to the assessment of the products of the different attributes. In general, it can be seen that products with the highest liking are placed on the right side of the graph (both pieces and burgers). For both chicken pieces and burgers, all liking attributes were close to each other, and were closest to the attributes juiciness and meaty flavour. These relations were also clear from the correlations found for these attributes. Liking of flavour was correlated with meaty flavour (*r* = 0.61, *p* < 0.01 for chicken analogue pieces; *r* = 0.58, *p* < 0.01 for burgers). However, flavour intensity was not important (Appendix A), indicating that liking is more related to the type of flavour and not its intensity. Liking of texture was mostly correlated with the mouthfeel attribute juiciness for chicken analogue pieces (*r* = 0.37, *p* < 0.01) and analogue burgers (*r* = 0.36, *p* < 0.01). The regression results indicate that juiciness is the most important mouthfeel attribute to focus on in meat analogue developments to create a meatlike texture, especially for burgers. For chicken analogue pieces, a higher score in juiciness did not necessarily relate to higher appreciation for the products. Moreover, higher juiciness did not lead to a greater texture similarity to cooked chicken meat (Appendix A), and the texture attributes fibrousness and cohesiveness also played a role. This may be related to the distinct fibrous structures found in cooked chicken meat. 

Our findings are in line with a previous work on consumer acceptance determinants for meat analogues, which found properties such as tenderness and juiciness to be the most important textural attributes for appreciation [52,58,59,60]. Other studies have also proposed that consumer liking for beef burgers is fuelled by lower hardness and chewiness [61]. However, in this study, we did not find any clear indication of such relationship. These results clearly indicate that it is important to control the juiciness of meat analogues, and that more understanding in these aspects is required.

### 3.6. Relationship between Composition and Sensory Properties

To understand the mechanisms behind the perception of the studied meat analogues, also correlations between compositional parameters and sensory properties were searched. As expected, the perception of a fibrouslike structure was not correlated to added fibres, and in general, no correlations between fibres and sensory textural attributes were found. Fibres can act as both a structural enhancer and a destabilizer, depending on the type of fibre [37,62]. Fibre origin was not specifically mentioned in the ingredients list of the studied samples and could thus not be taken into account in the analysis of our data. Furthermore, a small fibre size may impart a lower cohesiveness and explain a lower sensory fibrousness (Table 3).

With respect to fat-related attributes, perceived fattiness can be expected to be linked to total amount of fat, distribution of the fat, and fat release from the samples. When looking at just the fat content of the samples, chicken analogue pieces from the Vegetarische Slager (4.40 g/100 g fat) with less fat were experienced as more fatty than the Fry’s Strips (11.8 g/100 g fat) with a higher fat content. However, the first of these products had a remarkable fat absorption during preparation, whereas the latter lost fat during frying. The lowest scores for sensory fattiness were experienced for Quorn Pieces (Table 5), which can be explained by the low content of fat (2.8%), the negative fat absorption (Table 2 and Table 3), and the homogeneous structure (Figure 1C) in which the fat shows an even distribution of tiny fat particles. Samples with larger pools of oil received higher scores for fattiness. For example, the Vegetarische Slager—Mc2 Burger and the Beyond Meat—Beyond Burger presented larger oil pools in the CLSM images (Figure 3), and received one of the highest scores in fattiness perception. For these specific products, the original fat content was already relatively high, while the fat absorption during preparation was limited. Large pools of fat in meat analogues have already been linked to sensory fattiness [63]. On the other hand, similar fattiness scores were also attributed to the Linda McCartney Burger, having a fat content between those of the Vegetarische Slager—Mc2 Burger and the Beyond Meat—Beyond Burger and showing smaller oil pools and an almost even distribution of relatively large oil globules on CLSM (Figure 1J2). Therefore, not only the size of the oil pools but also other characteristics seem to influence fattiness perception.

Although fattiness did not show a correlation with fat content, sensory hardness decreased with the amount of fat (*r* = −0.710, *p* = 0.014), as well as the chewiness (*r* = −0.661, *p* = 0.027) of the analogue pieces. These parameters may also be related to the type of fat, but we did not specifically consider this in this study. For the burgers, no relationships between composition and sensory texture attributes were observed.

For all studied products, the perception of juiciness, which was found to be one of the most important attributes for liking, could not be related to composition. In general, a higher sensory juiciness is expected when a product contains more water or can expel more water during consumption. Therefore, properties such as expressible moisture or moisture content were expected to correlate with these attributes. However, for neither chicken pieces nor burger samples, juiciness was directly correlated with MC or EM (Table 7). These factors may also be related to the salt content of the samples. Higher salt content was linked to increased juiciness in chicken pieces (Table 7). Salt is known to induce solubilisation of proteins and enhance water-holding capacity (WHC) [64]. Higher salt content may therefore aid in holding more water, which may lead to higher water loss during compression. However, no relation between juiciness and EM was found for chicken pieces. In burgers, we did find a higher relation between juiciness and EM (−0.410), although not significant. However, no relation between juiciness and salt content was obtained. These results show that salt content may be relevant for dense structures, such as chicken analogue pieces, but is less relevant for less cohesive compositional products, such as burgers. In this case, the role of additives becomes more important. In addition to salt, other additives, such as methylcellulose and carrageenan, were used in various tested products. These thickening or gelling additives also affect WHC. In this study, we were not able to take these effects into account. The origin of juiciness in meat analogues can thus not simply be linked to measurable characteristics, but may arise from a combination of aspects related to both the composition and the structure of the samples. A better understanding of the link between the juiciness and structural aspects is thus necessary. This should be made with more model systems, in which certain structural features can be changed more controllably.

In chicken analogue pieces, salt was associated with a meaty flavour (*r* = 0.763, *p* = 0.006), while a negative correlation between these two parameters was observed in burgers (*r* = −0.700, *p* = 0.016) (Table 7). Analogue burgers contained on average equal concentrations of salt, but the meaty flavour, derived from the additional spices and flavourings, varied among samples. Different types of ingredients can be used in the meat analogue samples to achieve a meat flavour, such as yeast extracts, iron complexes, malt extracts, and various savoury flavourings and aromas [65,66,67,68]. Meaty flavour was best achieved in the Valess Pieces. This was the only sample containing milk protein, and meaty flavour notes have been identified in milk [69]. All other samples scored below 60 (on a 0–100 scale) on meaty flavour, confirming the difficulty of perceiving meaty flavours during consumption, because either ‘beany’ aftertastes were more dominant, or flavour release was inhibited by the binding of flavour compounds to the proteins within the network [70,71].

### 3.7. Relationship between Textural Properties and Sensory Properties

To understand to what extent the sensory profile could be linked to the measured textural attributes of the studied products, we investigated possible correlations between these aspects. The instrumentally measured hardness, chewiness, and cohesiveness of the chicken analogue pieces were not well correlated to the corresponding attributes in the sensory test (Table 7). Significant correlations were found only between moisture content and cohesiveness (0.776), fat content and hardness (−0.710), and fat content and chewiness (−0.661). This could indicate that for dense structures, such as analogue chicken pieces, the fat plays an important role for the texture. The lack of correlation for other attributes may be explained by the complexity and the variability of the structure of these product types. We already mentioned earlier that in the case of chicken analogue pieces, the fibrous nature, the presence of air pockets, and the distribution of fat led to an inhomogeneous structure. This was even more visible in the analogue burgers. In addition, for samples requiring extensive chewing, such as meat and meat analogues, the fact that TPA only simulates two bites may result in a limited representativeness of the obtained data for the entire mastication process. Sensory hardness is a relatively simple attribute that is mostly assessed at the first bite, but it has been shown that in chicken meat, hardness perception can increase after 10–12 bites [72]. Whether this also changes over multiple bites for meat analogues is not known. Attributes such as chewiness, springiness, and cohesiveness are much more complex attributes. For some sensory attributes, multiple physical measurements may be required to explain sensory perception [73]. Still, several products received scores for texture attributes similar to those of real chicken (Table 5). However, the liking scores of these samples (Table 6) were lower, which confirms the importance of (meaty) flavour for liking.

In contrast to chicken analogue pieces, TPA measurements for analogue burgers were predictive for sensory hardness (*r* = 0.857, *p* < 0.000), chewiness (r = 0.715, *p* = 0.004), and cohesiveness (*r* = 0.768, *p* = 0.001) (Table 7). However, no clear relation with fat content was obtained. This may suggest that in less cohesive structures, such as analogue burgers, fat may be less important for the textural properties. Although we did not see a clear link between the juiciness and texture properties for analogue chicken pieces, such relation was found for analogue burgers. A higher measured hardness was associated with a lower sensory juiciness (*r* = −0.557, *p* = 0.039). This suggests that for burgers, juiciness may be better controlled by textural parameters, and the results suggest that the hardness of the samples plays an important role. This may be related to the type of proteins and the processing conditions. Burgers based on wheat were perceived to be less hard (*r* = −0.632, *p* = 0.037). Additionally, higher measured cohesiveness correlated with sensory fibrousness (*r* = 0.545, *p* = 0.044), again indicating that the texture attributes were more closely related to each other than for analogue chicken pieces. Even though the values of both TPA and sensory evaluation were similar, the liking scores for analogue burgers were lower than those for the real beef burgers, indicating that resembling the textural characteristics of the burgers is not sufficient to increase consumer acceptance.

### 3.8. Relationship between Physicochemical Properties and Sensory Properties

The sensory properties of meat analogues are not just determined by the type of main protein used (e.g., soy, wheat, and pea), but more related to the specific protein structures as affected by different processing parameters. In the production of meat analogues, the aim is to obtain a fibrouslike structure resembling meat fibres. The more ordered the fibrous structure, the firmer the product [74]. No distinctive high sensory fibrousness was observed in AH Pieces Like chicken (Table 5), even though a clear fibrouslike structure was visible upon structural examination using CLSM (Figure 1). The sensation of fibrousness may thus also be related to the detection of such structures: if the fibrous network is relatively soft, it may not be perceived as fibrouslike in the mouth, as these structures can easily be broken down.

Juiciness was, next to meaty flavour, found to be the most important attribute for liking and perceived similarity to meat. According to Sha and Xiong (2020), the juiciness of meat analogues depends on the similarity between their fibrous protein network and the structure of animal muscle tissues. However, recently, Cornet and coworkers also showed that the presence of air pockets in meat analogues may be related to juiciness [38]. In our study, no direct correlation with measurable properties (such as expressible moisture), structural features (such as air pockets), or compositional factors (such as protein type) could be identified. It is therefore not yet clear what structural features determine juiciness in these products.

Although EM could not directly be linked to juiciness, a higher EM in analogue burger samples was associated with increased sensory hardness (*r* = 0.563, *p* = 0.045) and chewiness (*r* = 0.630, *p* = 0.021), while a higher amount of fat in the EM was associated with lower sensory hardness (*r* = −0.655, *p* = 0.015) and chewiness (*r* = −0.643, *p* = 0.018) (Table 7). The relation between EM and sensory hardness may be explained by the changes of the product during consumption. When more moisture is expelled from the product, the remaining structure becomes more dry and may be perceived as harder and chewier in comparison with the initial product. The fact that juiciness had a negative correlation with hardness (−0.557) could indicate that the softness of the product may be more related to juiciness than the released moisture from the product during consumption. This could be explained by the fact that softer samples fall apart more easily in smaller pieces, which would ease the consumption process. These results show that juiciness is not simply related to one physical parameter, and that it is a complex attribute linked to a combination of factors. More insights on juiciness and changes in juiciness in relation to structural changes during consumption are needed.

### 3.9. Consumer Habits and Attitudes

Large-scale Dutch food consumption studies indicate that the majority of consumers eat meat three to six times a week [7,75]. These numbers were confirmed in this study, as participants indicated that their main daily meal included meat about four to six times a week (37%) or two to three times a week (30%) (Figure 3). Meat replacers were consumed two to three times a week (27%) or less (34%). Thirty-seven percent of the participants indicated that they had tried meat replacers in the past, but did not consume them anymore (31%, past users), or did not consume meat replacers at all (6%, nonusers). There were no significant differences in consumption patterns between men and women.

The participants were asked about their motivation to either choose or avoid the consumption of meat-replacing products. Sustainability/environmental reasons were the main drivers to choose meat replacers (30%), counting more for men (32%) than women (28%) (*p* = 0.022). Other reasons were dietary variety (19%), taste (15%) (where more women (17%) liked the taste than men (12%)), and personal health reasons (13%). Price (24%) was identified as a main barrier for choosing meat replacers, followed by unappealing flavour (16%), and unappealing texture (12%) (Figure 4).

Of all the products, real meat was liked better than the meat analogues. Next to sensory reasons, liking is influenced by expectation, attitudes, habits, and familiarity [76,77]. In this study, habitual meat or meat replacer intake or attitudes towards meat replacers only marginally explained liking scores for both chicken analogue pieces and analogue burgers (2% or less). In the literature, lower liking of meat analogues by non(-frequent) users is attributed to a certain degree of food neophobia, and repeated exposure (increased familiarity) can increase the hedonic evaluation over time [26,78]. In contrast, in this study, nonusers gave higher liking scores compared with current or past users (*p* = 0.053 for chicken analogue pieces, *p* < 0.000 for analogue burgers). It could be that lack of previous experience lowers the sensory expectations, while past users may expect significant improvements. This suggests that ‘newbies’ to meat analogues may be more readily persuaded to consume these products when their features are liked. This should be investigated further. Increasing the sensory attractiveness of meat substitutes will increase the willingness of consumers to adopt a plant-based diet, even among meat-loving consumers [79]. Our study shows that for increasing willingness of consumption, liking of flavour may be more important than liking of texture and appearance.

### 3.10. Implications

The present findings have a series of implications for the focus of the development of new meat analogues. Manufacturers of meat analogues have aimed to imitate the structure and microstructure of meat as much as possible. Recent advances in texture development of meat replacers have already improved liking of meat replacers considerably [26]. Still, the sensory profile of about one-third of the chicken analogue pieces and one-third of the analogue burgers was unsatisfactory. Additionally, there were a substantial number of participants commenting about the aftertaste of products. In our study, we did not find clear correlations between the structure, textural attributes, sensory perception, and liking. However, our results suggest that flavour and juiciness are most important. From these results, it is not clear yet how juiciness is linked to the structure of meat analogues and how juiciness can be measured by experimental techniques. More insights into how structural aspects influence juiciness are thus required. Further exploration of relationships between composition, microstructure, textural, and sensory parameters is likely to benefit future developments of meat analogue products. In such studies, also the effect of taste and aftertaste should receive greater attention.

## 4. Conclusions

The aim of this study was to find relationships among composition, textural characteristics, sensory perception, and liking of meat analogues. With respect to the textural attributes of the different meat analogues, some observations could be explained by the structure. For example, lower hardness could be related to a broad distribution of small air pockets and a larger amount of fat. In addition, for extruded chicken analogue pieces, hardness and chewiness increased with the degree of fibrouslike structure in the product. However, most textural properties could not be clearly linked to the structure of the samples. Besides the parameters that we determined in our study, other factors may also be relevant, such as the water-holding capacity of the protein structures, the specific hardness of the fibrous areas, the size of the oil droplets, the oil-binding capacity. From the sensory evaluation, meaty flavour and juiciness emerged as the most important determinants for liking. Among all meat analogues in this study, meaty flavour was best achieved in a milk-based product. However, juiciness could not be explained by moisture content (MC), cooking loss (CL), or expressible moisture (EM). CLSM images showed that lower sensory juiciness and fattiness were perceived for meat analogues with a microstructure with many small fat globules, as opposed to larger fat globules or pools of oil. Although the fat itself does not seem to affect the released moisture, the distribution of fat may influence the hardness of the samples. These results, therefore, suggest that juiciness is a complex attribute and could be related to a combination of different compositional and textural aspects. In addition, it was also shown that texture parameters do not necessarily need to resemble those of real chicken or beef to obtain consumer appreciation. These results show that the development of meat analogues should focus on improving meaty flavour and juiciness. More research is needed to gain more insights into how these aspects can be linked to structural and textural parameters of meat analogues in order to control these aspects. Furthermore, the link between texture and sensory parameters should be investigated in more detail, as texture analysis and CLSM of commercially available meat analogues could only partly explain the sensory experience. As this study was conducted with commercial products that varied a lot in their composition, the contribution of these ingredients could not be investigated. The specific effects of such ingredients would need to be studied with model systems with a more controlled composition.

## Figures and Tables

**Figure 1 foods-11-02227-f001:**
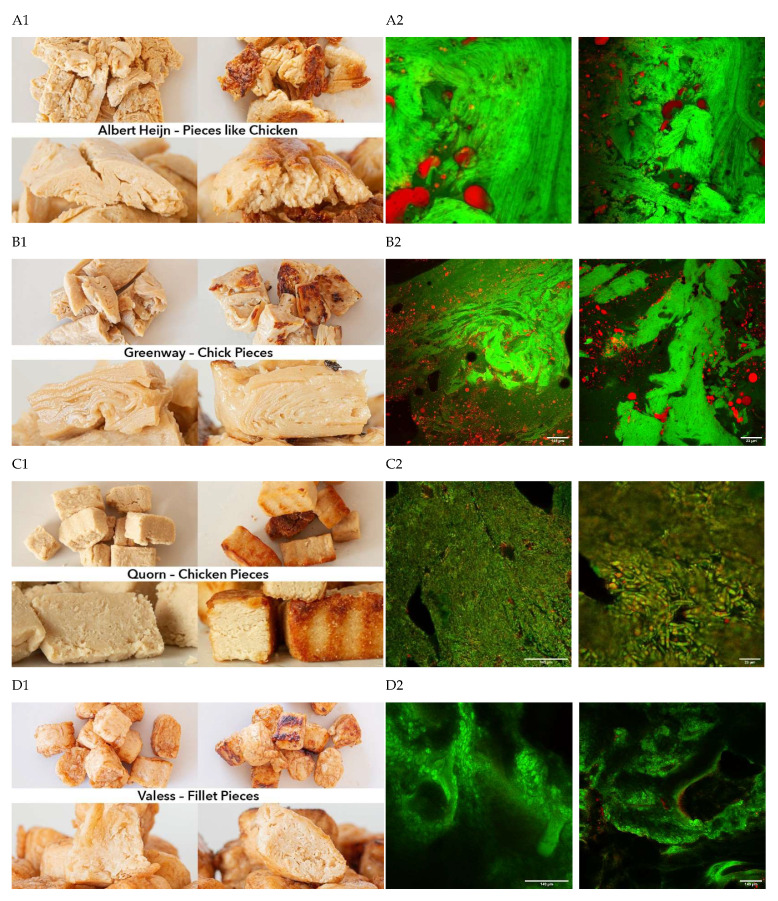
Top view and cross section of raw (left) and baked (right) products. CLSM. Pictures of the appearance of a selection of meat replacers (**A**–**L**) studied in the present research (left, 1) and CLSM micrographs of the same samples (right, 2). In the left column, the top 2 pictures are taken from the top of the products, while the lower pictures are taken from the front when the products were cut. In the right figure, representative CLSM images are given, with either a scale bar of 149 or 23 μm.

**Figure 2 foods-11-02227-f002:**
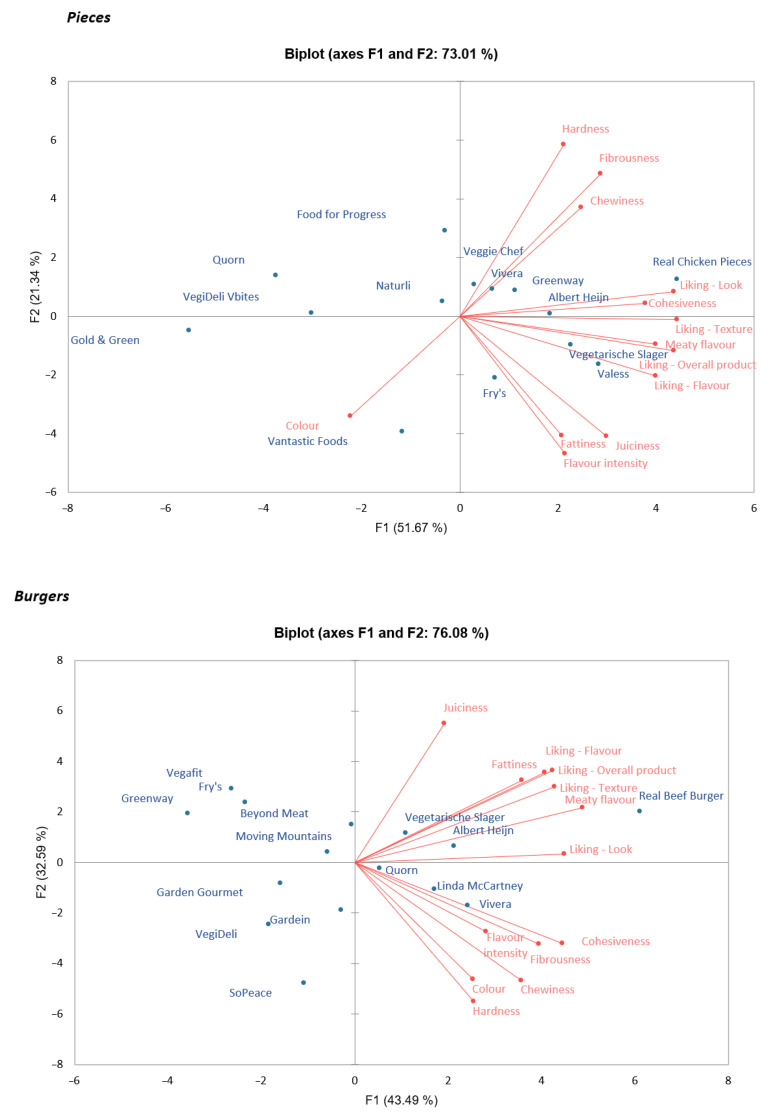
PCA plots for chicken pieces (**top**) and burgers (**bottom**). The different products are indicated in blue, and the attributes and liking are given in red. For chicken pieces, F1 relates to colour, liking and cohesiveness, and meaty flavour, and F2 related to most texture and mouthfeel attributes. For burgers, F1 relates to liking, meaty flavour, and fattiness, and F2 relates to juiciness, colour, and texture and mouthfeel attributes.

**Figure 3 foods-11-02227-f003:**
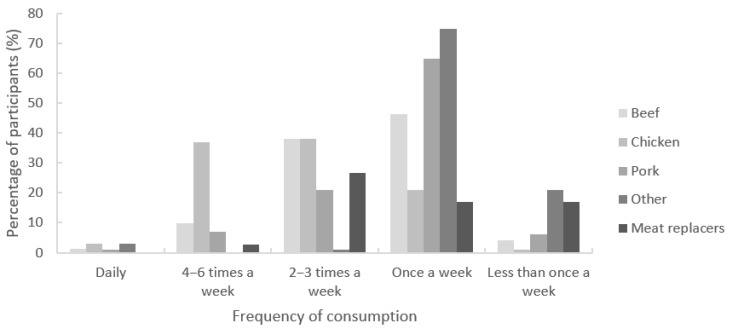
Frequency of meat and meat replacer (from left to right: beef, chicken, pork, other meats, meat replacers) consumption by the members of the sensory panel at the main meal of the day/dinner per type of meat (*n* = 71).

**Figure 4 foods-11-02227-f004:**
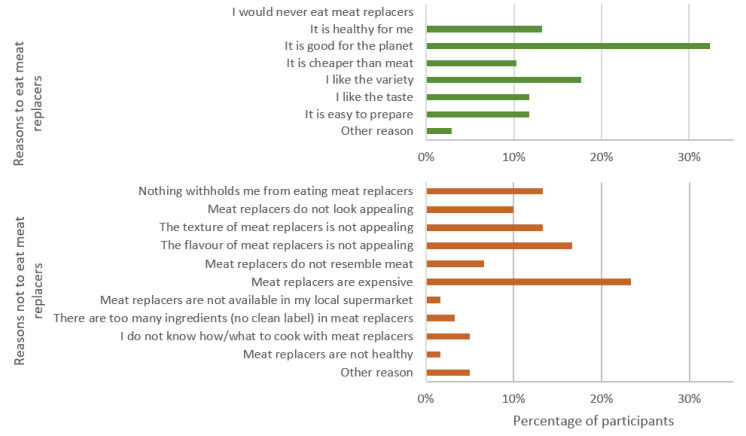
Reasons to (**top**) or not to (**bottom**) eat meat replacers (*n* = 71).

**Table 1 foods-11-02227-t001:** Attributes used in the sensory test and their description.

Attribute	Description
Colour	-
Hardness	Force required to compress the food between the molar teeth
Chewiness	Toughness of the products, the amount of work required to chew the product
Cohesiveness	Ease of fragmentation of the product upon mastication
Fibrousness	Detection of fibres
Juiciness	Amount of moisture released by the product during mastication
Fattiness	Degree of how fatty the product feels in the mouth
Flavour intensity	-
Meaty flavour	-

**Table 2 foods-11-02227-t002:** Selected meat analogue chicken pieces and beef burgers, with corresponding protein sources, protein content (%), fat content (%), fibre content (%), and moisture content (%).

Chicken Pieces	Protein Source	Protein (%)	Fat (%)	Fibre (%)	Moisture Content (%)	Burgers	Protein Source	Protein (%)	Fat (%)	Fibre (%)	Moisture Content (%)
Real Chicken Pieces (reference)	-	24	2.0	-	65.69 ^f^ ± 1.70	Ultimate Beef Burger (real beef)	-	20.0	17.0	0.0	59.29 ^ghi^ ± 0.62
AH—Pieces Like Chicken	Wheat (25%), pea (12%)	23.0	7.0	0.2	51.73 ^bcd^ ± 1.90	Albert Heijn —Burger Deluxe	Soy (53%), wheat (13%)	20.0	5.0	7.5	51.46 ^c^ ± 0.45
Food for Progress—Oumph the Chunk	Soy (23%)	17.0	0.4	5.1	64.94 ^ef^ ± 2.72	Beyond Meat —Beyond Burger	Pea, rice	18.0	19.0	2.6	53.67 ^cde^ ± 0.41
Fry’s—Chicken Style Strips	Wheat, soy	18.3	11.8	5.4	45.29 ^b^ ± 2.74	Fry’s—Traditional Burgers	Soy, wheat	14.0	5.6	6.2	59.93 ^hi^ ± 0.72
Gold & Green—Pulled Oats	Oat, pea (21%), fava bean (12%)	30.0	5.9	2.4	32.78 ^a^ ± 5.04	Gardein—Ultimate Beefless Burger	Soy, wheat, pea	18.8	5.2	3.1	59.98 ^hi^ ± 0.42
Greenway—Chick Pieces	Soy	17.8	2.6	7.3	55.86 ^cd^ ± 4.25	Garden Gourmet —Incredible Burger	Soy (19%), wheat	14.4	13.3	3.8	60.63 ^i^ ± 0.49
Naturli—Chick Free	Pea	21	1.7	2.8	49.83 ^bc^ ± 3.19	Greenway—Burger	Pea (22%), potato	14.0	10.0	-	56.57 ^efg^ ± 0.059
Quorn—Chicken Pieces	Mycoprotein (94%), egg	15.3	2.8	5.3	54.75 ^cd^ ± 1.47	Linda McCartney—Quarter Pounder	Soy (58%)	17.3	11.9	2.4	54.65 ^def^ ± 1.12
Valess—Fillet Pieces	Milk (76%), wheat	16.8	4.8	4.9	54.98 ^cd^ ± 0.66	Moving Mountains Veggie Burger	Pea, wheat, soy	15.3	17.6	5.8	53.30 ^cd^ ± 0.43
Vantastic Foods—Chicken-Style Pieces	Soy (35%), wheat (20%)	15.0	9.9	-	56.33 ^cd^ ± 0.34	Quorn—Supreme Vegan Burger	Wheat, pea, mycoprotein	21.0	14.0	3.1	39.77 ^a^ ± 2.97
Vegetarische Slager—Kipstuckjes	Soy (93%)	19.9	4.4	7.6	57.75 ^de^ ± 0.78	SoPeace—Burger	Pea (45%)	16.5	11.7	4.4	57.31 ^fgh^ ± 0.86
Veggie Chef—Kipstukjes	Soy (93%)	19.4	0.5	5.6	55.05 ^cd^ ± 0.73	Vegafit—Gehaktschijf	Wheat (38%), potato	15.5	15.6	4.5	45.98 ^b^ ± 0.64
VegiDeli VBites—Meat-Free Chicken Pieces	Wheat, soy	26.6	6.7	4.3	52.02 ^bcd^ ± 0.34	Vegetarische Slager—Mc2 No-Beef Burger	Soy, wheat	17.0	8.6	<0.5	51.68 ^cd^ ± 0.26
Vivera—Plant Chicken Pieces	Soy (93%)	19.0	0.5	5.6	54.26 ^cd^ ± 1.99	VegiDeli—Quarter Pounder	Soy	19.3	8.7	-	51.61 ^cd^ ± 1.82
						Vivera—Vegetable Burger Patty	Soy, wheat	17.0	5.4	6.0	54.58 ^cdef^ ± 0.59

Different letters refer to statistical differences for the different parameters in the column.

**Table 3 foods-11-02227-t003:** Instrumental properties measured for the meat analogues studied in the present research.

Chicken Pieces	Cooking Loss (%)	EM (g Fluid/g Sample) (%)	Fat in EM (%)	Fat Absorption (%)	Hardness (kPa)	Springiness	Cohesiveness	Chewiness (kPa)
Real Chicken Pieces	23.24 ^h^ ± 1.55	3.68 ^b^ ± 0.95	18.35 ^a^ ±5.12	0.042 ^cde^ ± 5.37	142.02 ^abcd^ ± 118.48	0.69 ± 0.033	0.40 ^ab^ ± 0.035	50.82 ^abcd^ ± 45.57
AH—Pieces Like Chicken	12.35 ^bcde^ ± 2.87	0.76 ^a^ ± 0.021	55.15 ^ab^ ± 12.19	−11.34 ^abc^ ± 7.20	86.06 ^abc^ ± 28.97	0.90 ± 0.022	0.65 ^ef^ ± 0.031	56.44 ^abcd^ ± 18.66
Food for Progress—Oumph the Chunk	17.38 ^efg^ ± 1.43	0.67 ^a^ ± 0.13	69.45 ^b^ ± 11.20	−15.37 ^ab^ ± 5.42	51.97 ^ab^ ± 49.20	0.72 ± 0.22	0.66 ^f^ ± 0.046	31.84 ^abcd^ ± 33.88
Fry’s—Chicken Style Strips	6.62 ^a^ ± 0.34	0.62 ^a^ ± 0.075	46.63 ^ab^ ± 31.17	−17.72 ^a^ ± 1.32	96.76 ^abc^ ± 30.66	0.83 ± 0.017	0.49 ^abcde^ ± 0.039	46.79 ^abcd^ ± 12.24
Gold & Green—Pulled Oats	10.80 ^abc^ ± 0.97	1.04 ^a^ ± 0.58	40.85 ^ab^ ± 29.43	9.27 ^e^ ± 3.27	72.70 ^abc^ ± 19.74	0.74 ± 0.029	0.40 ^ab^ ± 0.047	24.79 ^abc^ ± 6.90
Greenway—Chick Pieces	13.28 ^cdef^ ± 3.96	1.44 ^a^ ± 0.46	66.18 ^b^ ± 7.93	1.80 ^de^ ± 1.69	199.63 ^cd^ ± 46.89	0.75 ± 0.18	0.48 ^abcd^ ± 0.065	87.74 ^cde^ ± 8.46
Naturli—Chick Free	11.24 ^abcd^ ± 1.66	1.22 ^a^ ± 0.61	38.33 ^ab^ ± 30.00	24.21 ^f^ ± 5.85	24.65 ^a^ ± 16.25	0.88 ± 0.029	0.63 ^def^ ± 0.036	14.56 ^a^ ± 8.47
Quorn—Chicken Pieces	16.41 ^defg^ ± 0.68	1.17 ^a^ ± 0.15	66.83 ^b^ ± 4.75	−3.28 ^bcd^ ± 1.72	263.12 ^d^ ± 14.70	0.79 ± 0.023	0.36 ^a^ ± 0.016	93.63 ^de^ ± 3.96
Valess—Fillet Pieces	18.02 ^fgh^ ± 1.43	0.78 ^a^ ± 0.056	55.96 ^ab^ ± 5.39	−10.74 ^abc^ ± 3.42	127.22 ^abc^ ± 22.20	0.67 ± 0.062	0.43 ^abc^ ± 0.020	43.70 ^abcd^ ± 11.74
Vantastic Foods—Chicken-Style Pieces	7.95 ^ab^ ± 0.31	0.37 ^a^ ± 0.012	62.81 ^ab^ ± 2.22	4.06 ^de^ ± 1.87	25.82 ^a^ ± 7.55	0.91 ± 0.013	0.54 ^bcdef^ ± 0.043	16.37 ^ab^ ± 5.13
Vegetarische Slager—Kipstuckjes	18.91 ^gh^ ± 2.37	1.06 ^a^ ± 0.29	80.66 ^b^ ± 5.87	9.40 ^e^ ± 5.15	117.40 ^abc^ ± 35.75	0.65 ± 0.17	0.44 ^abc^ ± 0.12	48.78 ^abcd^ ± 34.51
Veggie Chef—Kipstukjes	9.81 ^abc^ ± 0.57	0.83 ^a^ ± 0.094	66.58 ^b^ ± 8.32	1.21 ^cde^ ± 0.73	170.84 ^bcd^ ± 36.83	0.83 ± 0.059	0.56 ^cdef^ ± 0.093	94.623 ^de^ ± 22.48
VegiDeli VBites—Meat-free Chicken Pieces	9.77 ^abc^ ± 0.42	0.46 ^a^ ± 0.054	81.22 ^b^ ± 1.30	4.52 ^de^ ± 1.99	148.51 ^abcd^ ± 56.34	0.81 ± 0.033	0.60 ^def^ ± 0.026	83.25 ^bcde^ ± 32.22
Vivera—Plant Chicken Pieces	11.40 ^abcd^ ± 1.64	0.54 ^a^ ± 0.054	65.68 ^b^ ± 3.80	−3.78 ^bcd^ ± 4.11	265.01 ^d^ ± 35.46	0.85 ± 0.042	0.52 ^abcdef^ ± 0.034	141.58 ^e^ ± 18.09
**Burgers**	**Cooking Loss (%)**	**EM (g Fluid/g Sample) (%)**	**Fat in EM (%)**	**Fat Absorption (%)**	**Hardness (N)**	**Springiness**	**Cohesiveness**	**Chewiness (N)**
Albert Heijn—Ultimate Beef Burger (real beef)	25.12 ^I^ ± 2.47	1.73 ^e^ ± 0.21	42.41 ^a^ ± 3.47	−0.96 ^bc^ ± 3.01	210.04 ^abc^ ± 14.39	0.73 ^bc^ ± 0.032	0.36 ^bcde^ ± 0.014	74.55 ^abc^ ± 5.29
Albert Heijn—Burger Deluxe	6.58 ^ab^ ± 1.04	0.45 ^ab^ ± 0.028	86.51 ^fg^ ± 0.46	3.67 ^c^ ± 0.64	179.80 ^ab^ ± 25.83	0.73 ^bc^ ± 0.042	0.33 ^bcd^ ± 0.019	55.28 ^ab^ ± 12.43
Beyond Meat—Beyond Burger	18.92 ^g^ ± 0.14	0.50 ^ab^ ± 0.018	86.69 ^fg^ ± 1.44	−5.31 ^ab^ ± 2.28	88.26 ^a^ ± 7.66	0.55 ^ab^ ± 0.010	0.31 ^abcd^ ± 0.031	18.95 ^a^ ± 1.40
Fry’s—Traditional Burgers	8.37 ^bc^ ± 0.26	0.52 ^ab^ ± 0.073	84.42 ^efg^ ± 2.46	−4.90 ^ab^ ± 1.05	181.04 ^ab^ ± 21.59	0.90 ^c^ ± 0.034	0.36 ^bcde^ ± 0.004	73.66 ^abc^ ± 10.78
Gardein—Ultimate Beefless Burger	23.18 ^hi^ ± 1.17	0.99 ^d^ ± 0.13	69.65 ^b^ ± 3.99	4.90 ^c^ ± 1.47	254.51 ^bcd^ ± 62.54	0.73 ^bc^ ± 0.077	0.51 ^e^ ± 0.011	122.24 ^bcd^ ± 38.69
Garden Gourmet—Incredible Burger	23.18 ^hi^ ± 1.17	0.77 ^bcd^ ± 0.060	81.85 ^defg^ ± 0.045	−8.85 ^a^ ± 1.92	156.44 ^ab^ ± 20.34	0.76 ^bc^ ± 0.059	0.42 ^cde^ ± 0.055	61.92 ^ab^ ± 17.64
Greenway—Burger	11.22 ^cde^ ± 0.084	0.39 ^a^ ± 0.014	87.32 ^fg^ ± 1.14	−0.98 ^bc^ ± 1.24	92.21 ^a^ ± 14.50	0.40 ^a^ ± 0.047	0.16 ^a^ ± 0.053	9.40 ^a^ ± 2.78
Linda McCartney—Quarter Pounder	13.84 ^ef^ ± 1.65	0.92 ^cd^ ± 0.22	73.39 ^bc^ ± 1.36	−0.21 ^bc^ ± 0.75	147.31 ^ab^ ± 47.50	0.67 ^abc^	0.35 ^bcd^ ± 0.073	46.74 ^a^ ± 20.18
Moving Mountains—Veggie Burger	15.23 ^f^ ± 0.88	0.40 ^a^ ± 0.082	87.40 ^fg^ ± 1.47	−5.49 ^ab^ ± 1.72	127.19 ^ab^ ± 25.76	0.64 ^abc^ ± 0.076	0.38 ^bcde^ ± 0.041	39.19 ^a^ ± 10.76
Quorn—Supreme Vegan Burger	7.00 ^ab^ ± 0.34	0.45 ^ab^ ± 0.16	88.20 ^g^ ± 4.30	−0.63 ^bc^ ± 1.29	199.78 ^ab^ ± 29.83	0.70 ^abc^ ± 0.019	0.37 ^bcde^ ± 0.033	65.52 ^ab^ ± 7.13
SoPeace—Burger	20.64 ^gh^ ± 0.71	0.61 ^abc^ ± 0.14	76.85 ^bcde^ ± 4.11	−1.80 ^abc^ ± 2.73	395.42 ^d^ ± 129.29	0.77 ^bc^ ± 0.063	0.47 ^de^ ± 0.018	193.73 ^d^ ±70.20
Vegafit—Gehaktschijf	6.80 ^ab^ ± 0.32	0.41 ^a^ ± 0.033	86.40 ^fg^ ± 0.84	4.40 ^c^ ± 0.77	69.81 ^a^ ± 2.15	0.55 ^ab^ ± 0.044	0.24 ^ab^ ± 0.020	11.78 ^a^ ± 1.57
Vegetarische Slager—Mc2 No-Beef Burger	10.37 ^cd^ ± 0.92	0.62 ^abc^ ± 0.013	80.28 ^cdef^ ± 0.69	1.88 ^bc^ ± 1.19	177.07 ^ab^ ± 81.03	0.62 ^abc^ ± 0.34	0.36 ^bcde^ ± 0.17	48.27 ^ab^ ± 28.97
VegiDeli—Quarter Pounder	4.23 ^a^ ± 0.48	0.35 ^a^ ± 0.012	87.66 ^fg^ ± 1.29	1.64 ^bc^ ± 7.68	175.10 ^ab^ ± 19.12	0.75 ^bc^ ± 0.077	0.28 ^abc^ ± 0.020	47.26 ^ab^ ± 11.37
Vivera—Vegetable Burger Patty	13.49 ^def^ ± 0.47	0.66 ^abcd^ ± 0.078	75.17 ^bcd^ ± 4.14	3.29 ^c^ ± 1.60	352.03 ^cd^ ± 45.19	0.69 ^abc^ ± 0.073	0.44 ^cde^ ± 0.007	143.43 ^cd^ ± 28.08

Note: Different letters in the same column are significantly different at *p* < 0.05. Absence of letters means no significant differences. Values are means of triplicate measures ± standard deviation. Different letters refer to statistical differences for the different parameters in the column.

**Table 4 foods-11-02227-t004:** Raw appeal and preparation scores (scales 1–5) for the meat analogues studied in the present research (*n* = 71). Different letters refer to statistical differences for the different parameters in the column.

Chicken Pieces	Raw Attractiveness (Look, Smell, Colour)	Looks Like Meat	Expect Good Taste	Preparation Is Easy	Preparation Takes Long	Preparation Like Meat
Real Chicken Pieces	3.89 (±1.04) ^f^	4.79 (±0.61) ^d^	4.24 (±0.75) ^f^	3.75 (±0.98) ^ab^	2.55 (±1.13) ^bc^	4.75 (±0.60) ^g^
Quorn—Chicken Pieces	2.33 (±1.03) ^ab^	1.82 (±1.02) ^a^	2.42 (±1.13) ^abc^	3.45 (±1.21) ^a^	2.65 (±1.32) ^c^	2.01 (±1.17) ^a^
Food for Progress—Oumph the Chunk	2.77 (±0.97) ^bcd^	3.01 (±1.21) ^c^	2.96 (±0.98) ^cde^	3.87 (±1.01) ^ab^	2.37 (±1.19) ^abc^	3.20 (±1.15) ^ef^
Fry’s—Chicken Style Strips	3.17 (±1.12) ^def^	2.32 (±1.14) ^ab^	3.25 (±1.14) ^de^	4.16 (±0.72) ^b^	1.84 (±0.93) ^a^	2.70 (±1.23) ^bcde^
Gold & Green—Pulled Oats	1.98 (±1.08) ^a^	2.10 (±1.27) ^a^	2.10 (±1.11) ^a^	3.43 (±1.24) ^a^	1.97 (±0.94) ^ab^	2.42 (±1.19) ^ab^
Greenway—Chick Pieces	2.66 (±0.97) ^bcd^	2.38 (±0.99) ^ab^	2.73 (±0.96) ^bcd^	3.77 (±0.90) ^ab^	2.01 (±0.92) ^ab^	2.96 (±1.01) ^bcde^
Naturli—Chick Free	2.50 (±1.06) ^bc^	2.37 (±1.22) ^ab^	2.52 (±1.04) ^abc^	3.96 (±0.90) ^ab^	1.77 (±0.83) ^a^	2.93 (±1.10) ^bcde^
Albert Heijn—Pieces Like Chicken	3.12 (±1.14) ^cde^	3.01 (±1.14) ^c^	3.24 (±1.10) ^de^	4.04 (±0.82) ^b^	1.87 (±0.95) ^a^	3.11 (±1.16) ^def^
Valess—Fillet Pieces	3.32 (±1.98) ^ef^	3.28 (±1.17) ^c^	3.35 (±1.11) ^e^	3.93 (±1.00) ^ab^	1.96 (±1.12) ^ab^	3.67 (±1.22) ^f^
Vantastic Foods—Chicken-Style Pieces	2.40 (±1.07) ^ab^	2.25 (±1.11) ^ab^	2.51 (±1.11) ^abc^	3.63 (±1.14) ^ab^	2.47 (±1.14) ^bc^	2.99 (±1.21) ^bcde^
Vegetarische Slager—Kipstuckjes	2.84 (±1.04) ^bcd^	2.77 (±1.05) ^bc^	2.99 (±1.06) ^cde^	3.82 (±0.90) ^ab^	2.19 (±0.95) ^abc^	3.07 (±1.15) ^cdef^
Veggie Chef—Chicken Pieces	2.48 (±1.01) ^b^	2.24 (±1.10) ^ab^	2.58 (±0.92) ^abc^	3.77 (±0.93) ^ab^	2.32 (±1.02) ^abc^	2.57 (±1.19) ^abcde^
VegiDeli VBites—Meat-Free Chicken Pieces	2.31 (±1.00) ^ab^	2.06 (±1.05) ^a^	2.35 (±0.97) ^ab^	3.81 (±1.12) ^ab^	2.14 (±1.13) ^abc^	2.49 (±1.11) ^abcd^
Vivera—Plant Chicken Pieces	2.45 (±0.97) ^ab^	2.03 (±1.03) ^a^	2.59 (±0.94) ^abc^	3.69 (±0.95) ^ab^	2.11 (±0.93) ^abc^	2.44 (±1.28) ^abc^
**Burgers**	**Raw Attractiveness (Look, Smell, Colour)**	**Looks Like Meat**	**Expect Good Taste**	**Preparation Is Easy**	**Preparation Takes Long**	**Preparation Like Meat**
Albert Heijn—Ultimate Beef Burger (real beef)	2.92 (±1.26) ^e^	4.34 (±1.01) ^d^	3.48 (±1.09) ^ef^	3.56 (±1.14) ^abc^	2.81 (±1.34) ^e^	4.68 (±0.65) ^e^
Albert Heijn—Burger Deluxe	3.62 (±0.97) ^f^	3.32 (±1.26) ^c^	3.82 (±0.78) ^f^	3.84 (±0.90) ^bc^	1.87 (±0.94) ^a^	2.83 (±1.20) ^abc^
Beyond Meat—Beyond Burger	2.37 (±1.14) ^abcd^	3.09 (±1.07) ^c^	2.61 (±1.22) ^abc^	3.23 (±1.24) ^a^	2.56 (±1.18) ^bcde^	2.71 (±1.16) ^abc^
Fry’s—Traditional Burgers	2.35 (±0.92) ^abcd^	2.08 (±0.94) ^a^	2.72 (±0.93) ^abcd^	3.76 (±0.97) ^abc^	2.30 (±1.12) ^abcde^	2.57 (±1.15) ^abc^
Gardein—Ultimate Beefless Burger	1.88 (±0.91) ^a^	1.73 (±0.83) ^a^	2.20 (±0.95) ^a^	3.69 (±1.02) ^abc^	2.04 (±0.97) ^abcd^	2.76 (±1.17) ^abc^
Garden Gourmet—Incredible Burger	2.53 (±1.19) ^bcde^	3.32 (±1.07) ^c^	2.73 (±1.16) ^abcd^	3.34 (±1.17) ^ab^	2.63 (±1.24) ^cde^	2.94 (±1.27) ^bcd^
Greenway—Burger	1.96 (±0.93) ^a^	2.04 (±0.97) ^a^	2.25 (±0.92) ^a^	3.53 (±1.03) ^abc^	2.41 (±1.12) ^abcde^	2.37 (±1.14) ^ab^
Linda McCartney—Quarter Pounder	2.25 (±0.99) ^ab^	2.28 (±1.10) ^ab^	2.47 (±0.99) ^ab^	3.80 (±0.99) ^abc^	1.87 (±0.87) ^a^	3.52 (±1.20) ^d^
Moving Mountains—Veggie Burger	2.32 (±1.06) ^abc^	2.77 (±1.09) ^bc^	2.62 (±1.01) ^abc^	3.32 (±1.28) ^ab^	2.67 (±1.27) ^de^	2.88 (±1.10) ^bcd^
Quorn—Supreme Vegan Burger	2.96 (±1.11) ^e^	3.35 (±1.11) ^c^	3.21 (±1.03) ^de^	3.68 (±0.89) ^abc^	2.31 (±1.18) ^abcde^	3.10 (±1.20) ^cd^
SoPeace—Burger	2.76 (±1.11) ^cde^	2.13 (±1.07) ^a^	2.90 (±1.15) ^bcde^	3.75 (±0.97) ^abc^	1.83 (±0.91) ^a^	2.75 (±1.24) ^abc^
Vegafit—Gehaktschijf	2.95 (±1.12) ^e^	2.14 (±1.05) ^a^	3.30 (±0.92) ^def^	4.00 (±0.94) ^c^	2.12 (±1.08) ^abcd^	2.17 (±1.26) ^a^
Vegetarische Slager—Mc2 No-Beef Burger	2.56 (±1.00) ^bcde^	2.18 (±1.06) ^ab^	2.90 (±1.04) ^bcde^	4.07 (±0.85) ^c^	1.94 (±0.95) ^ab^	2.84 (±1.23) ^abc^
VegiDeli—Quarter Pounder	2.85 (±1.27) ^de^	2.76 (±1.14) ^bc^	3.08 (±1.17) ^cde^	3.87 (±0.99) ^bc^	2.01 (±1.00) ^abc^	2.94 (±1.27) ^bcd^
Vivera—Vegetable Burger Patty	2.91 (±1.07) ^e^	2.30 (±1.05) ^ab^	3.09 (±0.91) ^cde^	3.97 (±0.88) ^c^	1.96 (±0.97) ^ab^	2.94 (±1.29) ^bcd^

**Table 5 foods-11-02227-t005:** Scores of the sensory attributes (scale: 0–100) assessed for the meat analogues studied in the present research (*n* = 71). Different letters refer to statistical differences for the different parameters in the column.

Chicken Pieces	Colour (Very Light–Very Dark)	Hardness (Very Soft–Very Hard)	Chewiness (Not–Very)	Cohesiveness (Not–Very)	Fibrousness (Not–Very)	Juiciness (Very Dry–Very Juicy)	Fattiness (Not–Very)	Flavour Intensity (Very Weak–Very Strong)	Meat Flavour (Not–Very)
Real Chicken Pieces	27.67 ^a^ ± 18.80	55.12 ^f^ ± 20.48	61.94 ^bcde^ ± 20.93	63.21 ^c^ ± 20.25	64.77 ^e^ ± 26.43	47.82 ^cd^ ± 25.61	36.00 ^bc^ ± 21.04	57.02 ^cde^ ± 22.38	88.37 ^g^ ± 15.87
AH—Pieces Like Chicken	47.65 ^bc^ ±18.06	48.27 ^cdef^ ± 16.66	59.32 ^bcde^ ± 17.72	54.33 ^bc^ ± 19.94	63.78 ^de^ ± 16.08	50.92 ^d^ ± 21.03	43.00 ^bcd^ ± 21.09	63.51 ^def^ ± 17.53	56.28 ^de^ ± 23.63
Food for Progress—Oumph the Chunk	33.25 ^a^ ± 16.35	53.48 ^f^ ± 20.61	66.92 ^e^ ± 20.00	59.32 ^bc^ ± 21.03	64.08 ^de^ ± 20.94	45.51 ^cd^ ± 21.95	41.81 ^bcd^ ± 25.73	37.41 ^a^ ± 25.40	28.45 ^ab^ ± 23.59
Fry’s—Chicken Style Strips	47.45 ^bc^ ± 18.51	38.42 ^bc^ ± 18.60	43.21 ^a^ ± 23.05	59.04 ^bc^ ± 20.75	35.42 ^ab^ ± 21.73	56.07 ^de^ ± 21.33	51.40 ^def^ ± 22.62	57.48 ^cde^ ± 20.31	48.88 ^cde^ ± 27.47
Gold & Green—Pulled Oats	94.22 ^f^ ± 5.18	40.56 ^bcd^ ± 26.04	45.06 ^a^ ± 26.52	29.42 ^a^ ± 24.31	39.16 ^bc^ ± 26.84	28.97 ^ab^ ± 21.95	35.49 ^bc^ ± 23.87	46.15 ^abc^ ± 25.84	28.46 ^ab^ ± 23.83
Greenway—Chick Pieces	48.93 ^bcd^ ± 17.96	55.55 ^f^ ± 17.29	62.80 ^de^ ± 17.59	58.05 ^bc^ ± 20.84	62.57 ^de^ ± 21.77	54.52 ^de^ ± 22.73	47.85 ^cde^ ± 23.99	57.45 ^cdef^ ± 19.90	41.89 ^bcd^ ± 25.88
Naturli—Chick Free	57.24 ^de^ ± 16.04	48.29 ^cdef^ ± 18.99	65.35 ^e^ ± 16.38	55.99 ^bc^ ± 19.99	56.09 ^de^ ± 21.22	48.00 ^cd^ ± 26.68	43.89 ^bcd^ ± 25.05	63.24 ^def^ ± 23.21	36.20 ^abc^ ± 25.84
Quorn—Chicken Pieces	30.16 ^a^ ± 17.24	40.92 ^bcde^ ± 23.05	44.98 ^a^ ± 26.12	49.73 ^b^ ± 27.12	39.46 ^bc^ ± 28.89	19.12 ^a^ ± 18.12	18.66 ^a^ ± 17.98	50.74 ^bc^ ± 24.43	31.02 ^ab^ ± 29.82
Valess—Fillet Pieces	56.48 ^cde^ ± 16.27	41.87 ^bcde^ ± 17.86	50.12 ^ab^ ± 19.64	60.60 ^bc^ ± 19.37	51.52 ^cd^ ± 22.16	56.91 ^de^ ± 20.38	43.51 ^bcd^ ± 18.88	66.23 ^f^ ± 17.76	71.61 ^f^ ± 20.21
Vantastic Foods—Chicken-Style Pieces	59.25 ^e^ ± 17.78	20.31 ^a^ ± 15.44	51.58 ^abcd^ ± 30.55	55.06 ^bc^ ± 27.39	25.42 ^a^ ± 22.94	72.15 ^f^ ± 20.40	60.50 ^ef^ ± 21.54	64.29 ^ef^ ± 20.00	40.43 ^abc^ ± 28.61
Vegetarische Slager—Kipstuckjes	52.37 ^bcde^ ± 16.79	46.87 ^bcdef^ ± 20.33	64.10 ^e^ ± 18.64	60.25 ^bc^ ± 18.57	62.38 ^de^ ± 20.54	66.10 ^ef^ ± 20.70	63.06 ^f^ ± 21.83	66.89 ^f^ ± 19.29	55.94 ^de^ ± 26.82
Veggie Chef—Kipstukjes	46.37 ^b^ ± 16.86	50.23 ^def^ ± 18.63	60.67 ^bcde^ ± 18.40	55.08 ^bc^ ± 21.04	55.88 ^de^ ± 21.78	49.59 ^d^ ± 21.48	44.40 ^bcd^ ± 24.75	43.71 ^ab^ ± 21.21	48.28 ^cde^ ± 27.42
VegiDeli VBites—Meat-Free Chicken Pieces	52.42 ^bcde^ ± 18.28	36.86 ^b^ ± 18.34	50.41 ^abc^ ± 25.79	54.84 ^bc^ ± 22.78	42.86 ^bc^ ± 24.55	36.21 ^bc^ ± 20.88	32.07 ^b^ ± 21.92	52.96 ^bcde^ ± 21.25	27.34 ^a^ ± 23.09
Vivera—Plant Chicken Pieces	51.22 ^bcde^ ± 19.28	51.71 ^ef^ ± 20.77	62.52 ^cde^ ± 19.90	58.08 ^bc^ ± 23.26	62.67 ^de^ ± 22.34	50.39 ^d^ ± 21.75	46.03 ^cd^ ± 23.48	51.62 ^bcd^ ± 21.25	47.42 ^cde^ ± 27.99
**Burgers**	**Colour (Very Light–Very Dark)**	**Hardness (Very Soft–Very Hard)**	**Chewiness (Not–Very)**	**Cohesiveness (Not–Very)**	**Fibrousness (Not–Very)**	**Juiciness (Very Fry–Very Juicy)**	**Fattiness (Not–Very)**	**Flavour Intensity (Very Weak–Very Strong)**	**Meat Flavour (Not–Very)**
Albert Heijn—Ultimate Beef Burger (real beef)	61.34 ^ef^ ± 14.83	52.57 ^cd^ ± 19.12	62.34 ^ef^ ± 17.73	62.82 ^d^ ± 18.25	56.14 ^b^ ± 25.71	74.73 ^g^ ± 19.66	69.18 ^f^ ± 20.17	69.52 ^cd^ ± 15.62	90.83 ^c^ ± 13.25
Albert Heijn—Burger Deluxe	58.52 ^def^ ± 16.54	46.64 ^bc^ ± 17.03	48.21 ^bc^ ± 19.38	57.76 ^bcd^ ± 19.85	51.29 ^b^ ± 20.57	50.69 ^de^ ± 20.12	45.71 ^abcde^ ± 19.21	64.50 ^bcd^ ± 18.77	51.90 ^b^ ± 26.27
Beyond Meat—Beyond Burger	52.12 ^cde^ ± 15.23	38.84 ^b^ ± 16.94	49.61 ^bcd^ ± 21.75	48.37 ^abc^ ± 23.44	52.54 ^b^ ± 25.18	55.82 ^def^ ± 20.37	53.47 ^de^ ± 21.95	52.65 ^a^ ± 21.12	53.21 ^b^ ± 29.18
Fry’s—Traditional Burgers	42.26 ^ab^ ± 19.26	27.01 ^a^ ± 18.21	33.39 ^a^ ± 23.56	44.93 ^ab^ ± 24.76	28.66 ^a^ ± 21.18	54.70 ^def^ ± 19.06	48.77 ^bcde^ ± 21.55	55.94 ^ab^ ± 21.52	42.12 ^ab^ ± 26.76
Gardein—Ultimate Beefless Burger	65.56 ^f^ ± 17.86	56.79 ^de^ ± 18.98	61.01 ^def^ ± 20.10	59.04 ^cd^ ± 22.80	46.12 ^b^ ± 26.35	37.99 ^abc^ ± 20.83	38.36 ^ab^ ± 22.62	58.60 ^abc^ ± 20.73	47.03 ^b^ ± 27.46
Garden Gourmet—Incredible Burger	47.65 ^bc^ ± 19.64	46.48 ^bc^ ± 20.97	54.07 ^bcde^ ± 20.69	54.42 ^abcd^ ± 21.54	51.40 ^b^ ± 25.25	48.02 ^cd^ ± 22.13	41.45 ^abcd^ ± 21.54	59.32 ^abc^ 2 1.99	41.90 ^ab^ ± 32.01
Greenway—Burger	46.43 ^bc^ ± 17.80	18.45 ^a^ ± 12.85	31.09 ^a^ ± 24.72	43.68 ^a^ ± 27.68	28.17 ^a^ ± 21.32	60.47 ^ef^ ± 20.46	48.13 ^abcde^ ± 21.76	60.60 ^abc^ ± 21.59	41.49 ^ab^ ± 25.89
Linda McCartney—Quarter Pounder	67.94 ^fg^ ± 14.40	58.10 ^de^ ± 18.79	60.72 ^cdef^ ± 22.13	54.34 ^abcd^ ± 22.54	57.15 ^b^ ± 23.31	51.24 ^de^ ± 18.97	51.08 ^cde^ ± 22.71	66.80 ^bcd^ ± 17.33	48.50 ^b^ ± 28.41
Moving Mountains—Veggie Burger	54.47 ^cde^ ± 17.25	40.04 ^b^ ± 18.39	49.17 ^bcd^ ± 23.09	55.60 ^abcd^ ± 23.33	50.72 ^b^ ± 23.53	63.20 ^fg^ ± 20.19	57.95 ^ef^ ± 19.50	57.60 ^ab^ ± 21.48	42.59 ^ab^ ± 28.91
Quorn—Supreme Vegan Burger	60.83 ^ef^ ± 16.79	43.84 ^bc^ ± 17.30	55.68 ^bcdef^ ± 20.84	55.83 ^abcd^ ± 21.39	57.99 ^b^ ± 22.85	54.78 ^def^ ± 19.19	52.54 ^cde^ ± 20.27	58.41 ^abc^ ± 20.55	48.74 ^b^ ± 26.85
SoPeace—Burger	61.26 ^ef^ ± 18.50	88.45 ^f^ ± 11.56	67.07 ^f^ ± 28.79	58.01 ^bcd^ ± 31.93	51.89 ^b^ ± 32.60	28.04 ^a^ ± 21.45	40.79 ^abc^ ± 23.70	69.52 ^cd^ ± 19.85	31.27 ^a^ ± 25.04
Vegafit—Gehaktschijf	35.44 ^a^ ± 20.58	17.83 ^a^ ± 15.70	27.54 ^a^ ± 24.47	44.79 ^ab^ ± 28.98	23.62 ^a^ ± 21.92	55.80 ^def^ ± 21.86	42.03 ^abcd^ ± 22.28	62.44 ^abcd^ ± 18.89	28.91 ^a^ ± 23.51
Vegetarische Slager—Mc2 No-Beef Burger	50.23 ^bcd^ ± 16.71	41.12 ^b^ ± 19.12	46.35 ^b^ ± 21.31	53.43 ^abcd^ ± 20.90	48.81 ^b^ ± 23.32	53.89 ^def^ ± 20.94	48.74 ^bcde^ ± 19.28	65.79 ^bcd^ ± 18.05	51.54 ^b^ ± 28.47
VegiDeli—Quarter Pounder	75.84 ^g^ ± 15.21	43.73 ^bc^ ± 19.45	47.43 ^b^ ± 23.82	49.48 ^abcd^ ± 25.54	46.02 ^b^ ± 27.28	33.33 ^ab^ ± 22.38	36.05 ^a^ ± 21.78	66.43 ^bcd^ ± 22.63	29.00 ^a^ ± 23.67
Vivera—Vegetable Burger Patty	64.31 ^f^ ± 15.55	65.22 ^e^ ± 16.72	62.63 ^ef^ ± 19.07	62.98 ^d^ ± 21.03	52.57 ^b^ ± 22.83	44.40 ^bcd^ ± 20.32	47.42 ^abcde^ ± 21.79	73.58 ^d^ ± 17.21	50.97 ^b^ ± 26.81

Note: Different letters in the same column are significantly different at *p* < 0.05. Values are means of 71 participant scores on a 0–100 VAS scale ± standard deviation.

**Table 6 foods-11-02227-t006:** Liking scores (scale 1–9), grade (scale 1–10), and intention to eat again (scale 0–2) for the meat analogues after preparation studied in the present research (*n* = 71).

Chicken Pieces	Look	Texture	Flavour	Overall Liking	Grade	Like to Eat It Again
Real Chicken Pieces	7.94 (±1.25) ^f^	7.99 (±1.04) ^h^	7.81 (±1.43) ^g^	7.94 (±1.09) ^g^	8.56 (±1.33) ^g^	1.89 (±0.40) ^f^
Albert Heijn—Pieces Like chicken	6.61 (±1.68) ^cde^	7.11 (±1.23) ^gh^	6.37 (±2.09) ^ef^	6.76 (±1.34) ^ef^	7.05 (±1.46) ^ef^	1.51 (±0.69) ^ef^
Food for Progress—Oumph the Chunk	6.79 (±1.64) ^de^	5.43 (±2.09) ^cde^	3.73 (±1.80) ^ab^	4.97 (±1.74) ^bc^	5.15 (±1.54) ^ab^	0.51 (±0.69) ^a^
Fry’s—Chicken Style Strips	6.32 (±1.82) ^cd^	6.63 (±1.61) ^fg^	6.83 (±1.93) ^fg^	6.79 (±1.60) ^ef^	7.10 (±1.73) ^ef^	1.45 (±0.75) ^de^
Gold & Green—Pulled Oats	4.11 (±2.44) ^a^	4.01 (±1.98) ^a^	3.49 (±1.93) ^a^	3.92 (±1.96) ^a^	4.27 (±1.82) ^a^	0.32 (±0.55) ^a^
Greenway—Chick Pieces	6.56 (±1.52) ^cde^	6.30 (±1.73) ^efg^	5.03 (±2.07) ^cd^	6.07 (±1.65) ^de^	5.98 (±1.71) ^bcd^	0.97 (±0.74) ^bc^
Naturli—Chick Free	6.54 (±1.69) ^cde^	5.21 (±2.15) ^bcd^	4.09 (±2.35) ^abc^	4.90 (±2.07) ^abc^	5.15 (±2.02) ^ab^	0.68 (±0.79) ^ab^
Quorn—Chicken Pieces	4.97 (±2.15) ^ab^	4.21 (±2.35) ^ab^	3.71 (±2.66) ^a^	4.13 (±2.15) ^ab^	4.46 (±2.02) ^a^	0.48 (±0.71) ^a^
Valess—Fillet Pieces	7.39 (±1.51) ^ef^	7.69 (±1.10) ^h^	7.59 (±1.02) ^g^	7.45 (±1.24) ^fg^	7.91 (±1.08) _fg_	1.87 (±0.38) ^f^
Vantastic Foods—Chicken-Style Pieces	5.68 (±2.19) ^bc^	4.83 (±2.49) ^abc^	4.89 (±2.46) ^bcd^	5.32 (±2.40) ^cd^	5.70 (±2.19) ^bc^	0.96 (±0.90) ^bc^
Vegetarische Slager—Kipstuckjes	6.55 (±1.52) ^cde^	6.38 (±1.81) ^efg^	5.97 (±2.10) ^def^	6.24 (±1.80) ^de^	6.73 (±1.65) ^de^	1.34 (±0.75) ^cde^
Veggie Chef—Chicken Pieces	Lso 6.14 (±1.45) ^cd^	6.06 (±1.80) ^def^	5.58 (±2.03) ^de^	5.87 (±1.56) ^cde^	6.29 (±1.55) ^cde^	1.11 (±0.78) ^cde^
VegiDeli VBites—Meat-Free Chicken pieces	4.96 (±2.00) ^ab^	4.42 (±2.05) ^abc^	3.93 (±2.09) ^abc^	4.30 (±1.88) ^ab^	4.69 (±1.84) _a_	0.51 (±0.77) ^a^
Vivera—Plant Chicken Pieces	5.72 (±1.73) ^bc^	6.30 (±1.77) ^efg^	5.46 (±2.12) ^de^	5.99 (±1.67) ^de^	6.20 (±1.51) ^cde^	1.08 (±0.77) ^cd^
**Burgers**	**Look**	**Texture**	**Flavour**	**Overall Liking**	**Grade**	**Like to Eat It Again**
Albert Heijn—Ultimate Beef Burger (real beef)	7.87 (±1.23) ^e^	7.58 (±1.59) ^g^	7.74 (±1.50) _f_	7.65 (±1.54) ^i^	8.44 (±1.43) ^h^	1.80 (±0.47) _h_
Albert Heijn—Burger Deluxe	7.11 (±1.32) ^de^	6.99 (±1.53) ^fg^	6.87 (±1.91) ^ef^	6.93 (±1.44) ^hi^	7.35 (±1.44) ^g^	1.58 (±0.65) ^gh^
Beyond Meat—Beyond Burger	5.17 (±2.20) ^ab^	4.46 (±1.79) ^efg^	5.83 (±2.21) ^cde^	5.80 (±1.99) ^defg^	6.53 (±1.79) _efg_	1.18 (±0.85) ^defg^
Fry’s—Traditional Burgers	5.72 (±1.80) ^bc^	4.93 (±2.01) ^bcd^	5.44 (±2.04) ^bcd^	5.53 (±1.83) ^cdefg^	5.93 (±1.65) ^cde^	0.92 (±0.84) _bcde_
Gardein—Ultimate Beefless Burger	5.20 (±2.10) ^ab^	5.59 (±1.75) ^cde^	4.89 (±2.21) ^abc^	4.23 (±1.85) ^bcde^	5.51 (±1.93) ^bcde^	0.83 (±0.81) ^bcd^
Garden Gourmet—Incredible Burger	4.44 (±2.03) ^a^	5.30 (±2.03) ^bcd^	4.23 (±2.55) ^ab^	4.61 (±2.17) ^bc^	5.27 (±2.22) ^bcd^	0.76 (±0.82) ^bcd^
Greenway—Burger	4.23 (±.190) _a_	4.27 (±2.06) ^b^	4.53 (±2.36) ^ab^	4.43 (±1.87) _ab_	5.09 (±1.93) _abc_	0.70 (±0.78) ^abc^
Linda McCartney—Quarter Pounder	6.63 (±1.50) ^cd^	5.99 (±2.04) ^def^	5.45 (±2.14) ^bcd^	5.90 (±1.77) ^efgh^	6.26 (±1.77) ^def^	1.06 (±0.81) ^cdef^
Moving Mountains—Veggie Burger	5.15 (±1.86) ^ab^	5.17 (±2.19) ^bcd^	4.41 (±2.36) ^ab^	4.80 (±2.18) ^bcd^	5.49 (±2.10) ^bcde^	0.80 (±0.87) ^bcd^
Quorn—Supreme Vegan Burger	5.63 (±1.83) ^bc^	5.87 (±2.23) ^cdef^	4.97 (±2.18) ^bc^	5.41 (±2.05) ^bcded^	6.07 (±1.93) ^cdef^	0.87 (±0.81) ^bcde^
SoPeace—Burger	5.90 (±1.99) ^bc^	2.65 (±2.09) ^a^	4.23 (±2.29) ^ab^	3.46 (±2.04) ^a^	4.13 (±2.07) ^a^	0.30 (±0.62) ^a^
Vegafit—Gehaktschijf	6.03 (±1.88) ^bc^	4.97 (±2.18) ^bcd^	6.24 (±2.22) ^de^	6.08 (±1.99) ^efgh^	6.28 (±2.09) ^def^	1.13 (±0.86) ^cdef^
Vegetarische Slager—Mc2 No-Beef Burger	6.35 (±1.56) ^cd^	6.45 (±1.80) ^ef^	6.49 (±2.01) ^de^	6.51 (±1.59) ^gh^	6.99 (±1.48) ^fg^	1.46 (±0.77) ^fgh^
VegiDeli—Quarter Pounder	6.00 (±1.73) ^bc^	4.86 (±1.97) ^bc^	3.69 (±2.17) ^a^	4.48 (±2.08) ^abc^	4.61 (±2.07) ^ab^	0.49 (±0.71) ^ab^
Vivera—Vegetable Burger Patty	6.31 (±1.72) ^cd^	5.93 (±1.87) ^cdef^	6.21 (±2.03) ^de^	6.30 (±1.64) ^fgh^	6.54 (±1.56) ^efg^	1.31 (±0.82) ^efg^

Different letters refer to statistical differences for the different parameters in the column.

**Table 7 foods-11-02227-t007:** Pearson correlation coefficients (r) between average sensory attributes, instrumental properties, and compositional parameters.

Chicken Pieces	Instrumental Properties						Nutritional Values		
Sensory Attribute	Hardness	Chewiness	Cohesiveness	Springiness	Moisture Content	%EM	Fat (g)	Fibres (g)	Salt (g)
Hardness	0.353	0.381	0.033	−0.165	0.253	0.497	−0.710 *	0.219	0.252
Chewiness	−0.108	0.191	0.475	0.360	0.566	0.423	−0.661 *	0.144	0.134
Cohesiveness	0.063	0.165	0.283	0.225	0.776 **	−0.053	−0.079	0.450	0.276
Fibrousness	0.182	0.238	0.163	−0.039	0.408	0.444	−0.578	0.086	0.269
Juiciness	−0.486	−0.075	0.323	0.508	0.319	−0.092	0.164	0.317	0.824 **
Fattiness	−0.503	−0.044	0.327	0.533	0.161	−0.034	0.163	0.336	0.825 **
Flavour intensity	−0.268	−0.355	−0.150	−0.168	−0.051	0.272	0.440	−0.075	0.575
Meaty flavour	0.033	−0.194	−0.401	−0.338	0.107	0.077	0.277	0.071	0.763 **
**Burgers**	**Instrumental properties**						**Nutritional values**		
**Sensory Attribute**	**Hardness**	**Chewiness**	**Cohesiveness**	**Springiness**	**Moisture content**	**%EM**	**Fat (g)**	**Fibres (g)**	**Salt (g)**
Hardness	0.857 **	0.837 **	0.827 **	0.823 **	0.325	0.563 *	−0.245	−0.033	−0.251
Chewiness	0.710 **	0.715 **	0.803 **	0.727 **	0.200	0.630 *	−0.134	−0.213	−0.575
Cohesiveness	0.668 **	0.687 **	0.768 **	0.668 **	0.104	0.533	−0.388	0.150	−0.599
Fibrousness	0.407	0.396	0.545 *	0.439	−0.121	0.361	0.099	−0.279	−0.721 *
Juiciness	−0.557 *	−0.513	−0.420	−0.424	−0.424	−0.410	0.418	0.025	−0.012
Fattiness	−0.173	−0.139	−0.062	−0.043	−0.369	−0.255	0.443	−0.024	−0.112
Flavour intensity	0.513	0.481	0.311	0.366	0.297	0.387	−0.565	0.300	−0.101
Meaty flavour	0.078	0.100	0.175	0.171	−0.135	0.299	−0.275	−0.185	−0.700 *

EM = expressible moisture (g fluid/g sample), * = significance up to *p* < 0.05, ** = significant up to *p* < 0.01.

## Data Availability

Data will be available upon request.

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
