# Peer review of "Meat Analogues: Relating Structure to Texture and Sensory Perception"

_foods, 2022, doi:10.3390/foods11152227_

Round 1
Reviewer 1 Report
This manuscript has attributed the sensory and textural properties of meat analogues to their protein type. However, there are many problems in the manuscript.
Hydrocolloids are key ingredients in meat analogues and have significant effects on the texture, microstructure, moisture content, water retention, oil holding capacity and sensory attributes of these products. However, the role of hydrocolloids such as xanthan, carrageenan, etc., in these products is not studied. On the other hand, usually meat flavor or flavor enhancers are added to meat analogues to improve their organoleptic properties. The type and concentration of these additives have dramatic effects on the taste and flavor of meat analogue but in this manuscript the taste and flavor of meat analogues are attributed to the source of protein. Additionally, the authors have reported correlations between the main protein type of the meat analogues and the fat content. But the differences in fat content are mainly due to the amount of oil added to these products and cannot be attributed to the source of proteins. The pore size and the size and distribution of fat globules is affected by several parameters such as mixing conditions, emulsifiers, stabilizers, fat content, the protein type, etc. Therefore, it is not possible to attribute this factor only to the source of protein.
Author Response
Reviewer 1
This manuscript has attributed the sensory and textural properties of meat analogues to their protein type. However, there are many problems in the manuscript.
Hydrocolloids are key ingredients in meat analogues and have significant effects on the texture, microstructure, moisture content, water retention, oil holding capacity and sensory attributes of these products. However, the role of hydrocolloids such as xanthan, carrageenan, etc., in these products is not studied.
We agree with the reviewer that hydrocolloids will indeed also change specific textural aspects. As we used commercial products, we do not have the specific information on the products to take these factors into account. To acknowledge that these factors are also important, we have added the following sentence in lines 515-517 as: “In addition to salt, other additives such as methylcellulose and carrageenan were used in various of the tested products. These thickening or gelling additives also affect WHC. In this study, we were not able to take these effects into account.”
To acknowledge this limitation of our study, we have added the following sentence at the end of the conclusion section in lines 666-669 as: “As this study was done with commercial products that varied a lot in their composition, the contribution of these ingredients could not be investigated. The specific effects of such ingredients would need to be studied with model systems with a more controlled composition.”
On the other hand, usually meat flavor or flavor enhancers are added to meat analogues to improve their organoleptic properties. The type and concentration of these additives have dramatic effects on the taste and flavor of meat analogue but in this manuscript the taste and flavor of meat analogues are attributed to the source of protein.
We agree with the reviewer that flavor is also important. As we used commercial products, we did not have information on these aspects. Therefore, we also did not look into these aspects further. The reviewer mentions that we attributed taste to the source of proteins. However, this is not really the case, and we did not link flavour to the protein source. We have checked the manuscript, but this link is never made. We did link flavor to liking, but this is indeed a result of different type of flavour aspects that arise from both the protein source and the added flavours. We have also acknowledged that flavour is an important aspect. In lines 524-527, we also acknowledge that flavour is different in the commercial products, and that this has an effect on the products: “Analogue burgers contained on average equal concentrations of salt, but the meaty flavour, derived from the additional spices and flavourings, varied among samples. Different types of ingredients can be used in the meat analogue samples to achieve a meat flavour, such as yeast extracts, iron complexes, malt extracts and various savoury flavourings and aromas.”
Additionally, the authors have reported correlations between the main protein type of the meat analogues and the fat content. But the differences in fat content are mainly due to the amount of oil added to these products and cannot be attributed to the source of proteins.
We agree with the reviewer that the fat content is not linked to the protein source. This is not what we intended to say. To clarify this aspects, we have added the following sentence in lines 219-222: “Although no direct link between protein type and fat content is expected, this may be related to the water-binding capacity of soy proteins [31,32], which is relatively high compared to other plant-based proteins. Higher water-binding may allow for less addition of fat to achieve a higher sensory juiciness.” We hope that this makes it more clear that we did not relate directly the protein type to fat content.
The pore size and the size and distribution of fat globules is affected by several parameters such as mixing conditions, emulsifiers, stabilizers, fat content, the protein type, etc. Therefore, it is not possible to attribute this factor only to the source of protein.
We again agree with the reviewer that the structure is related to many different parameters, and that links to protein source may be a coincidence. We have re-read the discussion section, but could not find the part that the reviewer refers to that we directly link pore size and fat distribution to protein source. We did mention that CL could be related to protein source. To avoid this confusion, we have rephrased the sentence as: “ In addition, no relation could be found between CL and type of proteins, protein content, content of added fibres, or other compositional factors. CL may thus be more related to specific features due to other compositional factors, i.e., emulsifiers, stabilizers, processing conditions, ingredient interactions and the properties of the formed protein network [37-39]” (Lines 260-263)

Reviewer 2 Report
This is a Well-written and relevant work that investigates the overall quality attributes and consumers' acceptability of plant based meats. Comments are attached for minor revisions.

Author Response
Reviewer 2
Adding salt traditionally is used to combat flavor, however, this influences nutrients (added Na+) and can impact protein structure (ionic influence). Reviewer recommends including this.
We thank the reviewer for pointing this out. To include that salt can also influence the structure, we have added the following sentence in lines 506-517: “These factors may also be related to the salt content of the samples. Higher salt content was linked to increased juiciness in chicken pieces (Table 7). Salt is known to induce solubilisation of proteins and enhance water holding capacity (WHC) [64]. Higher salt content may therefore aid in holding more water, which may lead to higher water loss during compression. However, no relation between juiciness and EM was found for chicken pieces. In burgers, we did find a higher relation between juiciness and EM (-0.410), although not significant. However, no relation between juiciness and salt content was obtained. These results show that salt content may be relevant for dense structures, such as chicken analogue pieces, but are less relevant for less cohesive compositional products, such as burgers. In this case, the role of additives becomes more important.”
The protein type will also influence the actual digestabilty. for example the PDCAAS of wheat is 0.25 vs soy which is close to 1. therefore the protein of plant proteins that is digestible (daily value) may be actually lower than the values reported. Was this considered? If so, it should be discussed.
In this study, we did not take into account the digestibility of the products, which was out of scope for this study. We do agree with the reviewer that this is an important aspects, and could be included in future studies.
Can this be discussed? Why does the author think soy burgers generally have lower fat? What is it about the function of soy that may influence this?
We do not know if there is a logical reason for this, or whether this is just a coincidence. However, it may be related to the water binding ability of the proteins. To discuss this, we have added the following sentence in lines 219-222: “Although no direct link between protein type and fat content is expected, this may be related to the water-binding capacity of soy proteins [31,32], which is relatively high compared to other plant-based proteins. Higher water-binding may allow for less addition of fat to achieve a higher sensory juiciness.”
Why are the true meet samples include for a reference point here?
We realized that the sentence was not clear, as we did not intend to refer to the real meat samples, but to the meat analogues samples. We have therefore rephrased the sentence into: “By taking samples from the middle of the products (as opposed to sampling from the edge), we assume the images reliably reflect the microstructure.” (lines 317-318). We hope that this description is more clear.
Easier to read by adding the statistical indicator. I.e. letters of the same letter among rows indicate statistically similar.
We thank the reviewer for pointing this out. However, the letters refer to differences on the parameters within a column. To make this more clear, we have added the following text in the caption of Table 4: “Different letters refer to statistical differences for the different parameters in the columns.”
Add statistical indicators by column
The statistical indicators were already added in the table, but apparently, this was not clear. To clarify this, we have added the following text in the caption of the table: “Different letters refer to statistical differences for the different parameters in the columns.”
Adding more details of the plot descriptions would enhance clarity. I.e. defining x and y axis in the legend.
We have added more information on the PCA plot in Figure 2: “The different products are indicated in blue, and the attributes and liking are given in red. For chicken pieces, F1 relates to colour, liking and cohesiveness, and meaty flavour, and F2 related to most texture and mouthfeel attributes. For burgers, F1 relates to liking, meaty flavour, and fattiness, and F2 relates to juiciness, colour, and texture and mouthfeel attributes.”
Add a key for protein type
We thank the reviewer for their remark. We have added a legend in figure 3 to make this more clear.
In addition to the amount of fat, did you also consider the type of fat as this too may influence the function.
As we do not have exact details on the composition, we did not take this into account. We agree with the reviewer that this is also an important factor. To make this more clear, we have added the following sentence: “These parameters may also be related to the type of fat, but we did not specifically consider this in this study.” in lines 498-499.
As mentioned above, sodium was not included above but would help with your narrative presented in this table as ther are some significance within Salt.
We thank the reviewer for pointing this out. As described above, we have no included a discussion on the relation with salt.

Reviewer 3 Report
Paper writing and conceptualization appear adequate for the purpose of the journal. Bibliographic sources appear adequate, and the topic is of interest for the scientific community and justifies the purpose of the research.
However, the paper appears unorganized and confused and, especially in the results section, in several cases the reader does not fully understand which specific test the authors refer to.
Here are some specific considerations:
- The participants in the sensory analyses are considered correctly as an untrained panel, but it must be specified that they are consumers who are informed about the characteristics of the product in question.
- Authors should indicate in materials and methods the reason that led to the choice of a 100-point scale and possibly provide supporting bibliography.
- The two PCA graphs, presumably related to sensory analysis, are not described or discussed in the text and the methodology used is completely absent from the materials and methods section.
In conclusion, the paper is of interest, but in the opinion of this reviewer there is the need for a thorough re-reading of the paper, making the meaning of the analyses clear in the results.
Author Response
Reviewer 3
Paper writing and conceptualization appear adequate for the purpose of the journal. Bibliographic sources appear adequate, and the topic is of interest for the scientific community and justifies the purpose of the research.
However, the paper appears unorganized and confused and, especially in the results section, in several cases the reader does not fully understand which specific test the authors refer to.
We thank the reviewer for the positive evaluation of our manuscript. We have tried to make it more clear by addressing the points made by the reviewer.
Here are some specific considerations:
- The participants in the sensory analyses are considered correctly as an untrained panel, but it must be specified that they are consumers who are informed about the characteristics of the product in question.
We thank the reviewer for pointing this out. We have already mentioned that the panel was a untrained panel in lines 146-147 as: “Quantitative sensory analysis was performed using a non-trained consumer panel (n=71, 63% female, 37% male), aged 19-41 years, and non-vegetarians.” To clarify that they were informed about the characteristics, we have added the following sentence in Line 161: “Prior to the tastings, participants were informed about the type of products (meat replacers) used in the test, and asked questions about their eating habits and attitudes regarding meat replacers.”
- Authors should indicate in materials and methods the reason that led to the choice of a 100-point scale and possibly provide supporting bibliography.
We thank the reviewer for pointing out this missing formation. We have added the following sentence and included a reference in Lines 167-170: “For the hedonic questions, a 9-point hedonic scale was used, as this scale has been reported to be reliable and have a high stability of response [28]. For the descriptive questions, a 0-100 VAS scale was used, as this scale allows for a high sensitivity of differences as panellists are not restricted by limited options [29].”
- The two PCA graphs, presumably related to sensory analysis, are not described or discussed in the text and the methodology used is completely absent from the materials and methods section.
We have added the following information in the M&M section in Lines 181-183: “The sensory data was further analysed using Principal Component Analysis (PCA) to identify factors that can explain most of the variance between the meat analogues. PCA was carried out using XLSTAT software (XLSTAT, 2022, Addinsoft, New York, NY).”
We have also discussed the PCA plot in the results & discussion section in Lines 446-454: “These findings were visualized using PCA-plots (Figure 2). The position of the tested products in the graphs (in blue) relates to the assessment of the products of the different attributes. In general, it can be seen that products with the highest liking are placed at the right side of the graph (both pieces and burgers). For both chicken pieces and burgers, all liking attributes are close to each other, and were closest to the attributes juiciness and meaty flavour. These relations were also clear from the correlations found for these attributes. Liking of flavour was correlated with meaty flavour (r=0.61, p<0.01 for chicken analogue pieces, and r=0.58, p<0.01 for burgers). However flavour intensity was not important (Supplementary information – Figure S3), indicating that liking is more related to the type of flavour and not its intensity.”
In conclusion, the paper is of interest, but in the opinion of this reviewer there is the need for a thorough re-reading of the paper, making the meaning of the analyses clear in the results.
We thank the reviewer for the positive feedback. We hope that the changes suggested by the reviewers make the paper easier to read. We have also added more discussions in the results and discussion section to make the meaning of the analyses more clear (Lines 390-392, 446-454, 467-468, 506-521, 539-541, 556-568, 598-599. We hope that these changes improved the manuscript.

Round 2
Reviewer 1 Report
The manuscript can be accepted
Reviewer 3 Report
The authors responded satisfactorily to all the observations made by this reviewer, and furthermore the readability of the article has considerably increased with the additions and modifications implemented. Therefore, in the opinion of this reviewer the article is now suitable for publication.